# Macrophages are exploited from an innate wound healing response to facilitate cancer metastasis

Tamara Muliaditan[1], Jonathan Caron[1], Mary Okesola[1], James W. Opzoomer[1], Paris Kosti[1], Mirella Georgouli[2], Peter Gordon[1], Sharanpreet Lall[1], Desislava M. Kuzeva[1], Luisa Pedro[3], Jacqueline D. Shields[3], Cheryl E. Gillett[1], Sandra S. Diebold[4], Victoria Sanz-Moreno [2], Tony Ng[1], Esther Hoste [5,6] & James N. Arnold [1]

Tumour-associated macrophages (TAMs) play an important role in tumour progression, which is facilitated by their ability to respond to environmental cues. Here we report, using murine models of breast cancer, that TAMs expressing fibroblast activation protein alpha (FAP) and haem oxygenase-1 (HO-1), which are also found in human breast cancer, represent a macrophage phenotype similar to that observed during the wound healing response. Importantly, the expression of a wound-like cytokine response within the tumour is clinically associated with poor prognosis in a variety of cancers. We show that co-expression of FAP and HO-1 in macrophages results from an innate early regenerative response driven by IL-6, which both directly regulates HO-1 expression and licenses FAP expression in a skin-like collagen-rich environment. We show that tumours can exploit this response to facilitate transendothelial migration and metastatic spread of the disease, which can be pharmacologically targeted using a clinically relevant HO-1 inhibitor.

[1] School of Cancer and Pharmaceutical Sciences, King's College London, Faculty of Life Sciences and Medicine, Guy's Campus, London SE1 1UL, UK. [2] Tumour Plasticity Laboratory, Randall Centre for Cell and Molecular Biophysics, King's College London, Guy's Campus, London SE1 1UL, UK. [3] Medical Research Council Cancer Cell Unit, Hutchison/Medical Research Council Research Centre, Cambridge CB2 0XZ, UK. [4] National Institute for Biological Standards and Control, Potters Bar, Hertfordshire EN6 3QG, UK. [5] Unit for Cellular and Molecular Pathophysiology, VIB Center for Inflammation Research, B-9052 Ghent-Zwijnaarde, Belgium. [6] Department of Biomedical Molecular Biology, Ghent University, Ghent B-9052, Belgium. These authors contributed equally: Tamara Muliaditan, Jonathan Caron. Correspondence and requests for materials should be addressed to J.N.A. (email: james.n.arnold@kcl.ac.uk)

Tumour-associated macrophages (TAMs) form part of the stromal cell infiltrate in solid tumours[1], and promote tumour progression through supporting angiogenesis[2], immune suppression[3], chemotherapeutic resistance[4–6] and tumour cell migration[7,8]. However, within the TAM population it has been demonstrated that there are phenotypic subsets with specific specialised roles[3,9,10]. A subpopulation of F4/80[hi] TAMs was identified in subcutaneous murine Lewis lung adenocarcinoma (LL2) tumours, which expressed surface fibroblast activation protein alpha (FAP) and intracellular haem oxygenase-1 (HO-1) and accounted for 10% of total F4/80[hi] cells[3]. FAP is a dipeptidyl peptidase capable of degrading gelatin and type I collagen[11,12], and also has a role in cellular signalling in cancer-associated fibroblasts (CAFs)[13]. HO-1 is an inducible enzyme responsible for the breakdown of haem to generate biliverdin, ferrous iron and carbon monoxide (CO)[14]. Selective conditional ablation of the FAP[+] TAM population in an immunogenic ovalbumin (OVA)-expressing LL2 tumour using diphtheria toxin in bone marrow chimeric FAP/diphtheria toxin receptor transgenic (DTR Tg) mice, resulted in an immunological control of tumour growth demonstrating that this macrophage subset played an important role in immune suppression[3,15]. FAP[+] TAMs represented the major tumoural source of HO-1 and pharmacological inhibition of this enzyme paralleled the observations made with conditional depletion of the producing cells, suggesting that HO-1 was essential to their biological function within the tumour[3]. As FAP[+] TAMs can also be found in human breast tumours[16], it is important to elucidate the full biological implications of this TAM subset, as well as their origin.

Macrophages are one of the most plastic cells of the immune system and display an exquisite ability to respond to environmental cues which shape their phenotype[17]. The biological responses of these cells are exploited by the tumour to drive progression of the disease. In the current study, we highlight the ability of the tumour to orchestrate a microenvironment which phenocopies the cytokine milieu and extracellular matrix of a superficial wound. As a result, the macrophages are coerced to instigate a wound healing response, identified by co-expression of FAP and HO-1, exemplifying Harold Dvorak's seminal observation 40 years ago that cancers resemble 'wounds that do not heal'[18]. This study demonstrates that tumours exploit the innate regenerative response of macrophages to facilitate metastatic spread of the disease.

## Results

### FAP[+] HO-1[+] TAMs represent a tumour-educated phenotype.
Selective conditional ablation of FAP[+] HO-1[+] TAMs, or pharmacological inhibition of their HO-1 activity, results in a cessation of tumour growth in subcutaneously implanted immunogenic LL2/OVA tumours, suggesting that these cells are a non-redundant population within the tumour microenvironment, and that HO-1 expression might represent a key effector molecule in their pro-tumorigenic functions[3]. As FAP[+] TAMs have been demonstrated to reside in human mammary adenocarcinoma[16], we investigated whether these cells within the human tumour microenvironment could also express HO-1. Indeed, FAP[+] HO-1[+] CD11b[+] myeloid cells could be found in tissue sections of human mammary adenocarcinoma (Fig. 1a, b), indicating that this phenotype is conserved across murine and human tumours. These FAP[+] HO-1[+] cells co-expressed the myeloid marker CD14 (Supplementary Figure 1a), suggesting that they are TAMs. To gain biological insight into the origin of these cells in breast cancer, we utilised an orthotopic model of mammary adenocarcinoma in which 4T1 tumour cells are injected into the mammary fat pad of Balb/c syngeneic mice[19]. The F4/80[+] TAMs

present in these tumours represented $10.8 \pm 3.3\%$ of all live tumoural cells (Fig. 1c) and expressed FAP, alongside the macrophage/TAM markers CCR2, CD11b, CD14, MHCII, IL4-R and MMR, with a low expression of the dendritic cell marker CD11c (Fig. 1d and Supplementary Figure 1b). CD11b[+] Ly6C[hi] monocytes, which had yet to differentiate to TAMs, did not express FAP (Fig. 1e), suggesting that FAP represented a differentiation marker. The expression of FAP by the macrophages was a stable phenotype in this model, and was constant during the growth of 4T1 tumours (Fig. 1f and Supplementary Figure 1c). Expression of tumoural *Hmox1*, the gene encoding HO-1, was also detected at the point of tumour establishment (day 9 post inoculation) (Fig. 1g and Supplementary Figure 1c) and was associated with the TAM population, where its expression correlated with FAP (Fig. 1h). The FAP[+] TAM phenotype was not a biological response to the microenvironment of the mammary fat pad in which the tumour cells were injected as TAMs in another orthotopic murine model of mammary adenocarcinoma, E0771[20,21], grown at the same anatomical location in syngeneic C57Bl/6 mice, did not express FAP (Fig. 1i) and expressed lower levels of *Hmox1* (Fig. 1j). Tumours have been demonstrated to expand populations of myeloid cells, which are functionally distinct from inflammatory monocytes[22–24]. 4T1 tumour cells, when injected into mice, expand peripheral populations of myeloid cells primarily through the secretion of granulocyte colony-stimulating factor (G-CSF)[25] (Supplementary Figure 1d), a phenomenon which has also been observed in the clinic[26]. To exclude the possibility that the FAP[+] TAM population represents a predefined differentiation state, G-CSF was ectopically expressed in E0771 cells and these cells were injected into C57Bl/6 syngeneic mice, resulting in an expansion of peripheral myeloid cell populations (Supplementary Figure 1e, f). However, this did not result in the presence of FAP[+] macrophages in these tumours (Supplementary Figure 1g), indicating that FAP[+] TAMs are not directly derived from this G-CSF-induced myeloid population. To exclude any effect of strain differences in the TAM response[27], 4T1 and E0771 cells were concurrently injected into opposing mammary fat pads of Balb/c *Rag2*[−/−]-immunocompromised mice (syngeneic to the 4T1 cells in which FAP[+] HO-1[+] TAMs had been identified). At day 21 post inoculation, both tumours were excised (Supplementary Figure 1h) and their respective TAM phenotypes assessed. In this setting, FAP[+] TAMs, which highly expressed *Hmox1*, were exclusively populating the 4T1 and not the E0771 tumours (Fig. 1k, l). These data collectively prompted the conclusion that FAP[+] HO-1[+] TAMs represent a tumour-educated phenotype rather than an autonomous peripherally derived population.

### FAP[+] HO-1[+] macrophages are associated with a wound response.
Having established that the FAP[+] macrophage phenotype was a response to the microenvironment, we considered whether any population of tissue-resident macrophages might express FAP during homoeostasis. Therefore, CD45[+] F4/80[+] macrophages from healthy tissues were analysed for FAP expression. FAP expression could not be detected on CD45[+] F4/80[+] tissue macrophages populating the liver, brain, lung, kidney, heart, spleen or visceral adipose (Supplementary Figure 2a). This suggested that the FAP[+] phenotype might have represented a specific response programme of these cells. To gain further biological insight into the FAP[+] HO-1[+] macrophage phenotype, TAMs were sorted from 4T1 tumours and analysed by microarray for changes in gene expression relative to basal M-CSF-differentiated splenic-derived macrophages (Fig. 2a). Gene ontology enrichment analysis of the transcriptome of these cells revealed a collection of immune-response pathways, including

pathways relating to the wound healing response (Fig. 2b). As tumours and healing wounds share many similarities[18], it prompted the consideration that the expression of FAP and HO-1 by macrophages might represent part of a wound healing response programme of these cells. We therefore analysed the abundance of FAP[+] HO-1[+] macrophages in healthy and wounded skin of mice 2 days (acute injury inflammatory response phase) and 6 days (wound healing response phase) after full-thickness dorsal skin wounding by a punch biopsy[28–31] (Fig. 2c, d and Supplementary Figure 2b). Indeed, FAP[+] HO-1[+] macrophages could be found in the granulation tissue at both 2 and 6 days post wounding (pw, Fig. 2d, e and Supplementary Figure 2b). FAP[+] HO-1[+] macrophages accounted for 14 ± 8% of F4/80[+] cells at day 2 pw and were maintained at an equal prevalence at day 6 pw (Fig. 2f, g). As expected, F4/80[+] macrophages accumulated at the wound edge between days 2 and 6 pw[32] (Fig. 2f); however, there was no increased prevalence of FAP[+]

HO-1[+] macrophages at this site (Fig. 2g), suggesting that FAP[+] HO-1[+] macrophages were a phenotype associated with the granulation tissue.

These observations suggest that the FAP[+] HO-1[+] macrophages in tumours share similarities to those of an innate wound healing response, potentially being exploited by the tumour.

**4T1 tumours phenocopy a wound cytokine response.** As we established that the FAP[+] HO-1[+] macrophage phenotype can be detected during the reparative inflammation phase of the wound response of the dorsal skin, we assessed the transcriptome of a murine wound at the same site[28] for potential secreted molecules which might direct macrophage differentiation and polarisation. In the acute inflammatory phase of the wound healing response, *Spp1*, *Il1b*, *Csf3*, *Il6*, *Osm* and *Il24* were significantly upregulated (fold change >16; ANOVA, $P < 0.05$) relative to unwounded skin

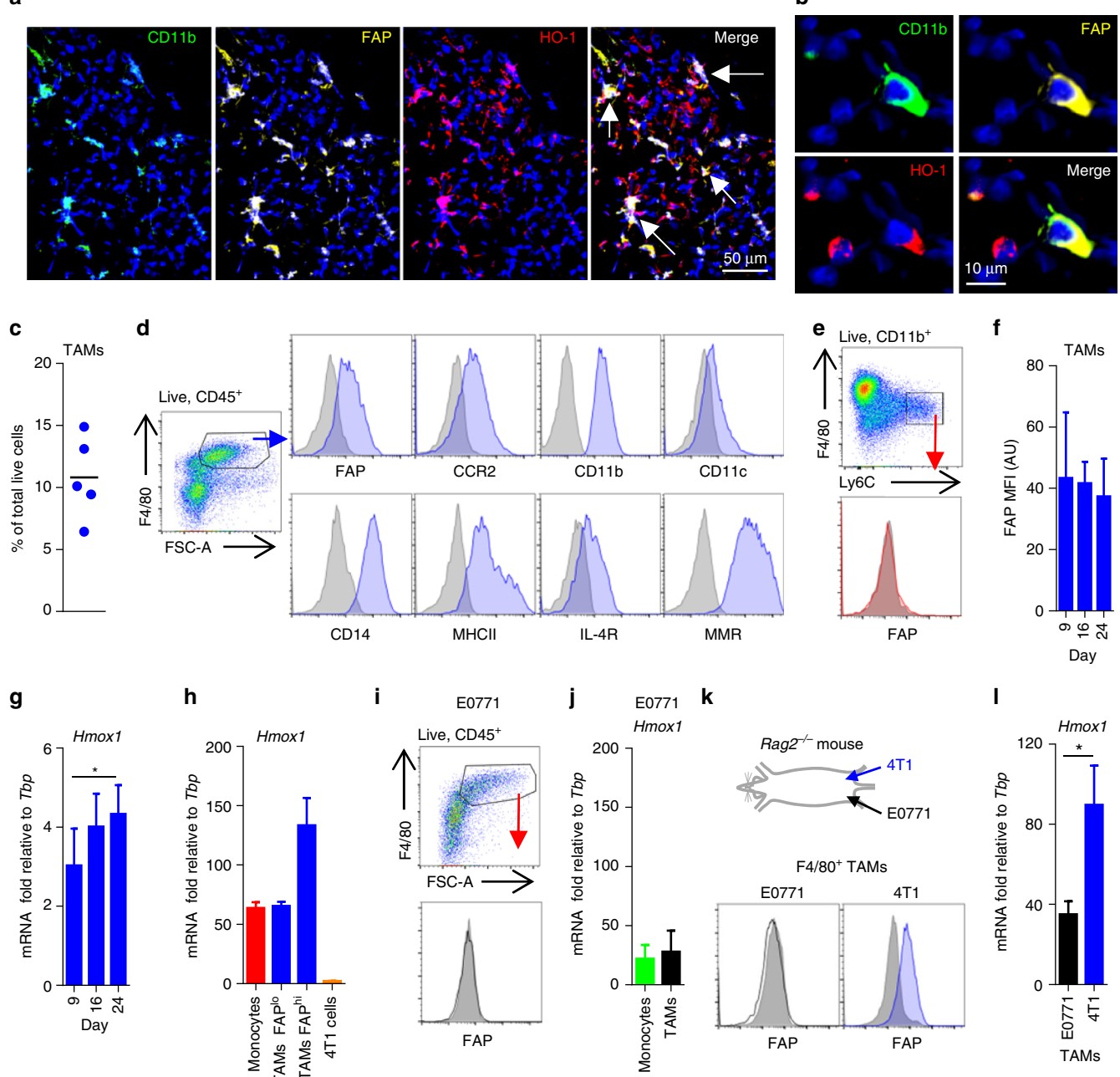

(Fig. 2h), with the peak of the cytokine response occurring at 24 h pw (Fig. 2i). The relative peak of the inflammatory response to wounding coincided with the identification of FAP⁺ HO-1⁺ macrophages in the granulation tissue by confocal microscopy analysis at day 2 pw (Fig. 2g). The expression levels of these cytokines were assessed in 4T1 tumours, which revealed a striking similarity to the reparative tissue response, including significantly higher *Spp1*, *Il1b*, *Csf3*, *Il6* and *Osm* expression in 4T1 tumours compared to healthy mammary fat pad tissue (Fig. 2j). This wound-like cytokine mileu was not detected in E0771 tumours which were devoid of FAP⁺ TAMs (Fig. 2j), demonstrating that this cytokine environment was specific to the 4T1 model and associated with the presence of FAP⁺ TAMs. Expression of *Il4*, *Il10* and *Il13*, which represent cytokines commonly associated with alternative macrophage activation, was undetectable in 4T1 tumours (Fig. 2j). No difference in expression levels of *Il24* was observed between 4T1 and E0771 tumours (Fig. 2j), which was expected since IL-24 is a known tumour suppressor gene[33], inhibiting cell proliferation and migration in the wound healing context and thus preventing tumorigenesis[34]. Analysis of the microarray data for the FAP⁺ TAM population isolated from 4T1 tumours (Fig. 2a) revealed that these cells were also directly expressing some of the acute wound inflammatory response genes, including *Il1b*, *Il6* and *Osm*, alongside the phenotypic markers *Hmox1* and *Fap* (Fig. 2k). Interestingly, these TAMs, as a collective population, also displayed characteristics of both alternative (*Arg1*, *Ym1*, *Il10* and *Fizz1*) and classical (*Nos2* and *Tnf*) macrophage activation (Fig. 2k), consistent with that previously described for both TAMs and wound healing macrophages[35].

**Healing wound cytokines indicate poor prognosis in cancer.** To investigate whether the inflammatory cytokine response observed in the murine models was conserved among species we investigated a human wound transcriptome data set. Several of the genes observed to be upregulated in the murine models, including *IL1B*, *IL6* and *SPP1*, were also highly over-expressed after thermal injury of human skin[36] (Supplementary Figure 3a) and in superficial wounds resulting from human skin grafting[37] (Supplementary Figure 3b). We therefore defined the species-conserved healing wound cytokines (HWCs) as *SPP1*, *IL1B* and *IL6*, and considered whether the expression levels of these genes might have prognostic value in human cancers. Interestingly, high expression of HWCs in breast cancer was associated with poor

prognosis (Fig. 3a). This association was not specific for breast cancer, but also observed in lung, gastric and ovarian cancer (Fig. 3b–d). These observations indicate that tumours can create a cytokine microenvironment associated with tumour progression, which is similar to that of an acute inflammatory wound healing response.

**IL-6 regulates HO-1 expression in macrophages.** To investigate the effect of the murine inflammatory wound response cytokines on the macrophage phenotype, bone marrow (BM) cells were incubated for 72 h with M-CSF to induce macrophage differentiation in the presence or absence of the respective cytokines, and then subsequently analysed for *Hmox1* gene expression. As a comparator, macrophages were also stimulated with IL-4, a cytokine commonly associated with the TAM phenotype[7,38]. IL-6 stimulation evoked a striking upregulation of *Hmox1* expression (Fig. 4a). However, IL-6 alone did not account for the wider prognostic value of the HWCs in the Kaplan–Meier plotter data set (Supplementary Figure 3c), but, IL-6 has been demonstrated to be associated with poor overall and disease-specific survival (DSS) by others[39]. The ability of IL-6 to regulate HO-1 was not specific to BM-derived macrophages (BMDM), as IL-6 also upregulated HO-1 in splenic Ly6C^hi monocyte-derived macrophages (Fig. 4b and Supplementary Figure 4a). In contrast, alternative (IL-4) and classical (IFNγ/LPS) macrophage activation programmes resulted in less HO-1 expression than IL-6 stimulation (Fig. 4b and Supplementary Figure 4a). The IL-6-HO-1 axis was also not species specific as IL-6 significantly upregulated HO-1 in human CD14⁺ monocyte-derived macrophages (Fig. 4b, c and Supplementary Figure 4b). 4T1 tumours express high levels of *Il6* mRNA (Fig. 2j), and the protein could be detected in the serum (Fig. 4d). As such, we investigated the tumoural source of IL-6 in sorted populations of cells from 4T1 tumours and identified that *Il6* expression was restricted to the stroma in this model, with the TAMs expressing the highest levels (Fig. 4e). Collectively, these data suggest that HO-1 expression by macrophages is a characteristic of the IL-6 polarisation programme.

**Collagen licences IL-6-dependent FAP expression.** We investigated whether IL-6 could also regulate FAP expression in BMDM; however, IL-6 alone was insufficient to induce FAP expression on these cells (Fig. 4f). This suggested that a secondary signal might be required. FAP is capable of degrading type I collagen[11], which is a major extracellular matrix (ECM) protein of the dermal layer

**Fig. 1** FAP⁺ HO-1⁺ TAMs in human and mouse breast adenocarcinoma are a tumour-educated phenotype. **a**, **b** Representative images of frozen human invasive ductal mammary carcinoma sections stained with DAPI (nuclei; blue) and antibodies against CD11b (green), FAP (yellow) and HO-1 (red) (representative images from *n* = 3 tumours). Arrows indicate cells co-expressing CD11b, FAP and HO-1. **c**, **d** Abundance (**c**) and surface characterisation (**d**) of live (7AAD⁻) CD45⁺ F4/80⁺ TAMs from enzyme-dispersed 4T1 tumours (at day 24 post inoculation) as assessed using flow cytometry. Each point in **c** represents live F4/80⁺ cells in an individual tumour. Histograms represent positive staining for the markers shown (blue shaded) against that of the respective isotype controls (grey shaded). **e** Flow cytometry gating strategy for live CD11b⁺ F4/80^low Ly6C^hi monocytes in a representative enzyme-dispersed 4T1 tumour; histogram shows surface FAP staining (red shaded) against that of the isotype control (grey shaded). **f** Median fluorescence intensity (MFI) of FAP surface expression with fluorescence minus one (FMO) background staining subtracted on CD45⁺ F4/80⁺ TAMs from enzyme-dispersed 4T1 tumours on the indicated day post injection of the tumour cells (*n* = 6). **g**, **h** *Hmox1* mRNA expression relative to the housekeeping gene *Tbp* in 4T1 tumour tissue on the indicated day post inoculation of the tumour cells (*n* = 6–8 tumours per time point) (**g**) and FACS-sorted monocytes, FAP^lo F4/80^hi TAMs and FAP^hi F4/80^hi TAMs and 4T1 tumour cells (CD45⁻ Thy1.2⁻ CD31⁻) from duplicate wells from a 4T1 tumour, data representative of duplicate experiments (**h**). **i** Representative FAP staining of live (7AAD⁻) CD45⁺ F4/80^hi TAMs from an enzyme-dispersed E0771 tumour as assessed using flow cytometry. Histogram represents positive staining for FAP (black line) against isotype control staining (grey shaded). **j** *Hmox1* mRNA expression relative to the housekeeping gene *Tbp* in FACS-sorted monocytes (CD11b⁺ F4/80^lo Ly6C^hi) and TAMs (F4/80^hi) from duplicate wells from an E0771 tumour, data representative of duplicate experiments. **k**, **l** 4T1 (blue) and E0771 (black open) tumours were grown concurrently in opposing mammary fat pads in *Rag2⁻/⁻* mice. **k** Schematic representation of the experimental setting (top) and representative FAP expression as assessed using flow cytometry on live CD45⁺ F4/80^hi TAMs from enzyme-dispersed tumours (bottom). **l** *Hmox1* mRNA expression relative to the housekeeping gene *Tbp* in MACS-sorted TAMs from enzyme-dispersed tumours (*n* = 4). Bar charts represent mean + s.d. *P < 0.05

of the skin that can play an important role in shaping cutaneous cellular responses[40]. We therefore considered whether FAP expression might be a response to the collagen-rich environment. To investigate if type I collagen could induce FAP expression in BMDMs, macrophages differentiated in the presence or absence of IL-6 were plated onto a 3D type I collagen matrix. This resulted

in transient IL-6-dependent induction of FAP expression (Fig. 4g and Supplementary Figure 4c), suggesting that collagen can provide a contextual location signal to these cells, shaping their response to IL-6. To date, the only transcription factor known to be critical for expression of the *Fap* gene is EGR1[41]. In agreement, alongside FAP expression, there was a concomitant induction of

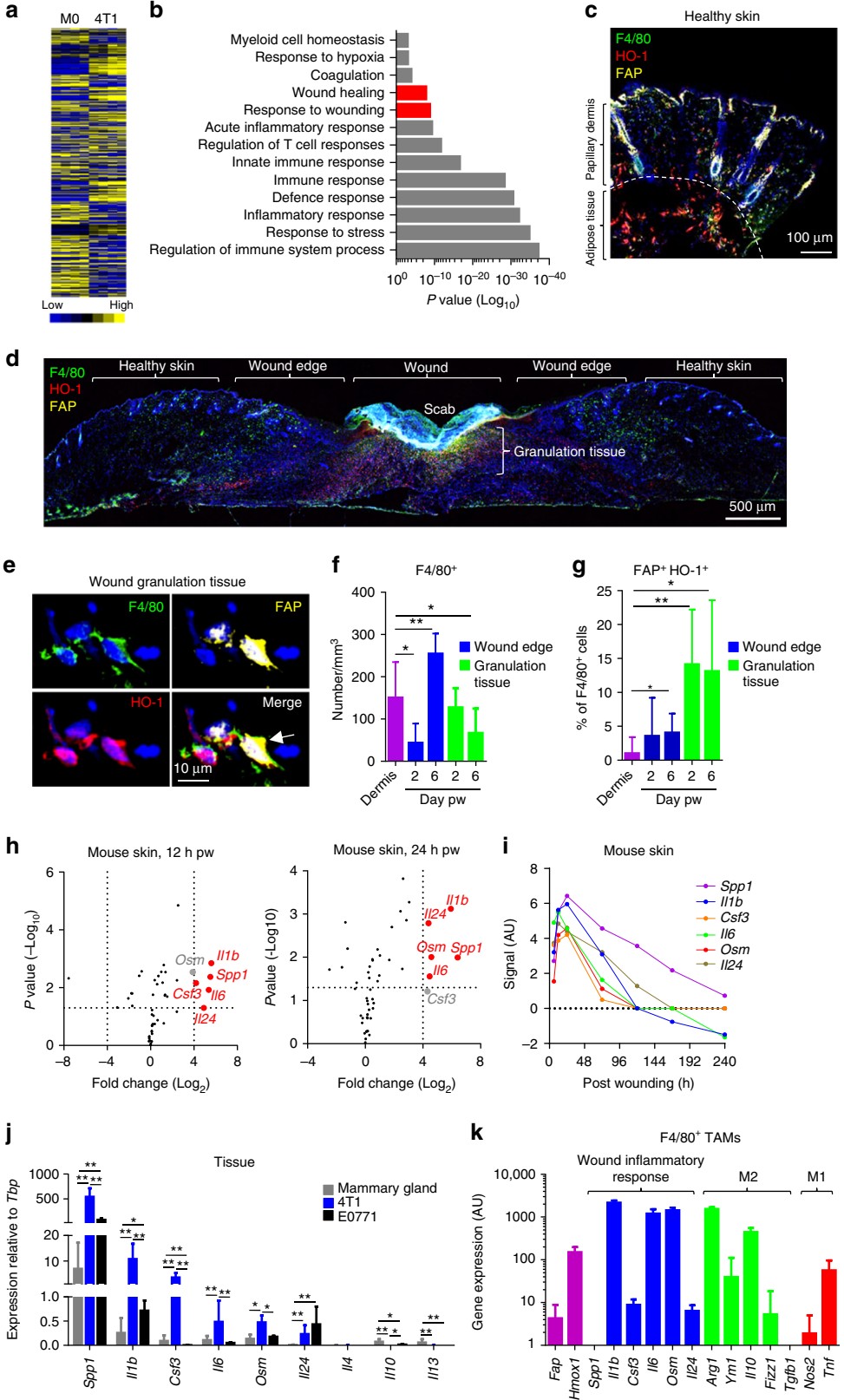

EGR1 expression in IL-6-stimulated BMDMs plated onto a collagen matrix (Fig. 4g and Supplementary Figure 4c). To extrapolate these in vitro findings back to our in vivo models, fibrillar collagen was analysed in healing wounds and tumours by second-harmonic generation (SHG) imaging. A dense fibrillar collagen network could be detected in healing wounds which was also present in healthy skin (Fig. 4h and Supplementary Figure 5a) as well as in 4T1 but not E0771 tumours (Fig. 4i, j and Supplementary Figure 5a). The ECM was strikingly different between 4T1 and E0771 tumours (Supplementary Figure 5b). In particular, expression of *Col1a1*, the gene for type 1 collagen and the substrate for FAP, was significantly higher in the 4T1 tumours (Fig. 4k). There was no inter-tumour correlation of the TAM's expression of FAP and tumoral *Col1a1* expression (Supplementary Figure 5c). However, broadly the expression of *Col1a1* in 4T1 tumours was equivalent to that found in the skin (Supplementary Figure 5d). As both IL-6 and collagen are highly abundant in healing wounds, these findings suggest that tumours can mimic a wound-like microenvironment which maintains an FAP[+] HO-1[+] TAM phenotype.

**IL-6 regulates the FAP[+] HO-1[+] TAM phenotype in 4T1 tumours**. To investigate whether IL-6 is capable of orchestrating the FAP[+] HO-1[+] TAM phenotype in 4T1 tumours in vivo, 4T1 cells were injected into syngeneic Balb/c WT and *Il6[−/−]* mice (Fig. 5a). The tumours grew equally well in the absence of IL-6, and at day 23 tumours were excised and analysed. In 4T1 tumours grown in *Il6[−/−]* mice, there was a modest increase in the proportion of TAMs (Fig. 5b). However, in agreement with the in vitro observations (Fig. 4g), in the absence of IL-6, the TAMs expressed significantly less FAP (Fig. 5c) and fewer had detectable HO-1 (Fig. 5d, e and Supplementary Figure 6a). These data confirm that IL-6 is important in shaping the TAMs towards an innate wound healing macrophage phenotype. Interestingly, not all TAMs in WT 4T1 tumours had detectable HO-1, however this might be explained by differences in the differentiation state of these cells, as both monocytes and TAMs express F4/80 (Fig. 1d, e), and HO-1 expression correlated with FAP expression in the TAM population (Fig. 1h). As FAP expression was associated with EGR1 upregulation (Fig. 4), we assessed the presence of nuclear EGR1 in TAMs in 4T1 tumours grown in WT and *Il6[−/−]* mice and found a significant reduction of nuclear EGR1 in the TAMs of *Il6[−/−]* mice (Fig. 5f, g). Both expression (Supplementary Figure 6b) and nuclear translocation (Supplementary Figure 6c) of EGR1 were reduced in F4/80[+] TAMs in *Il6[−/−]* tumours, suggesting that loss of nuclear EGR1 expression might be responsible for the reduction of FAP expression by TAMs in the absence of IL-6. FAP is more widely associated with a population of CAFs in the tumour microenvironment[15]. In the absence of IL-6 there was a small, but significant reduction in Thy1[+] CAF

abundance (Supplementary Figure 6d, e); however, their surface expression of FAP was unaffected (Supplementary Figure 6d, f), suggesting that the IL-6-dependent FAP expression observed in the TAM population was a cell-specific response. IL-6 has also been shown to regulate collagen expression[42], however, *Col1a1* (Fig. 5h) as well as other ECM proteins expressed in 4T1 tumours (Supplementary Figure 5b) *Col3a1*, *Tnc* and *Fn1*, were unaffected by the loss of IL-6 (Supplementary Figure 6g–i). These data demonstrate that IL-6 is a key regulator of the FAP[+] HO-1[+] macrophage wound healing response phenotype.

**FAP[+] HO-1[+] TAMs are associated with metastasis**. Having established that the FAP[+] HO-1[+] TAM phenotype was regulated by IL-6 signalling, we next considered whether these cells might also populate specific anatomical locations within the 4T1 tumour, and discovered a tendency for these cells to accumulate around viable vasculature (Fig. 6a, b and Supplementary Figure 7a). Since tumour cells and macrophages have been demonstrated to be closely associated during the intravasation event across the endothelium[43,44] and epithelial cell proliferation and migration is a crucial part of the wound healing response[45], we considered whether FAP[+] HO-1[+] macrophages might facilitate metastatic spread. We therefore treated mice bearing established 4T1 tumours with a clinically relevant HO-1 inhibitor, tin mesoporphyrin (SnMP)[3,46,47]. SnMP treatment did not affect primary tumour growth (Fig. 6c), but significantly suppressed pulmonary metastasis when analysed at day 24 post inoculation of tumour cells (Fig. 6d). The composition of the stroma in 4T1 tumours remained largely constant over time, with the most pertinent changes being a steady increase in the fraction of neutrophils (Supplementary Figure 7b, c), which have been implicated in the metastatic cascade[48], and a slight but steady decrease in the fraction of TAMs and CAFs. However, inhibition of HO activity did not affect the abundance of myeloid cells in these tumours (Fig. 6e). Lung metastasis-associated macrophages (MAMs)[10] and neutrophils[48] have been demonstrated to play key roles in the establishment of pulmonary metastasis. However, MAMs were distinct from the TAMs as they did not express FAP (Fig. 6f, g), and SnMP treatment did not affect MAM or neutrophil numbers in the lung (Fig. 6h, i). In order to elucidate whether HO activity was facilitating the intra- or extravasation event, 4T1-eGFP cells were injected i.v. into mice receiving SnMP. Abrogation of HO-1 activity did not compromise the ability of 4T1 cells to colonise the lung (Fig. 6j), suggesting that SnMP might have targeted the intravasation event at the primary tumour site rather than extravasation and colonisation in the lung. As IL-6 was a regulator of the FAP[+] HO-1[+] TAM phenotype in this model, we analysed pulmonary metastasis in 4T1 tumours grown in *Il6[−/−]* mice, and in agreement, found significantly fewer metastases in the absence of host-derived IL-6

**Fig. 2** FAP[+] HO-1[+] macrophages represent a wound healing response phenotype. **a** Heat map of raw gene expression values from splenic-derived M-CSF basal macrophages (M0) and 4T1 TAMs (*n* = 4 per condition). **b** Gene ontology enrichment analysis of M0 vs. 4T1 TAMs, showing selected pathways of interest that were significantly enriched. **c–e** Skin sections stained using DAPI (nuclei, blue) and antibodies against F4/80 (green), FAP (yellow) and HO-1 (red). **c** Image of healthy skin taken using a ×40 objective. **d** Image of wounded skin taken using a ×4 objective 6 days post wounding (pw). **e** High magnification image of example F4/80[+] FAP[+] HO-1[+] cells found in the granulation tissue; arrow marks cell of interest. **f, g** Quantification of F4/80[+] cells normalised to area (**f**) and percentage of F4/80[+] cells that co-expressed FAP and HO-1 (**g**) as an average of multiple fields of view in the healthy dermis and in the respective area of the wound environment (*n* = 6 mice per condition). Statistics were performed to compare each condition to healthy dermis. **h** Volcano plots showing changes in cytokine gene expression in skin in response to wounding using a 1 mm punch biopsy needle[28]. Genes marked in red represent cytokines most highly induced 12 h (left) and 24 h (right) pw (*P* ≤ 0.05; Log$_2$ (fold change) ≥4). **i** Expression levels of the indicated genes (**h**) over the duration of the wound response to a 1 mm punch biopsy needle[28]. **j** mRNA expression of the indicated cytokine genes relative to the housekeeping gene *Tbp* in healthy mammary gland (*n* = 6) and late stage (>20 day post inoculation) 4T1 (blue) and E0771 (black) tumours (*n* = 6). **k** M0-subtracted gene expression from 4T1 TAMs, displaying selected genes of interest (*n* = 4). Bar charts represent mean + s.d. *P < 0.05, **P ≤ 0.01

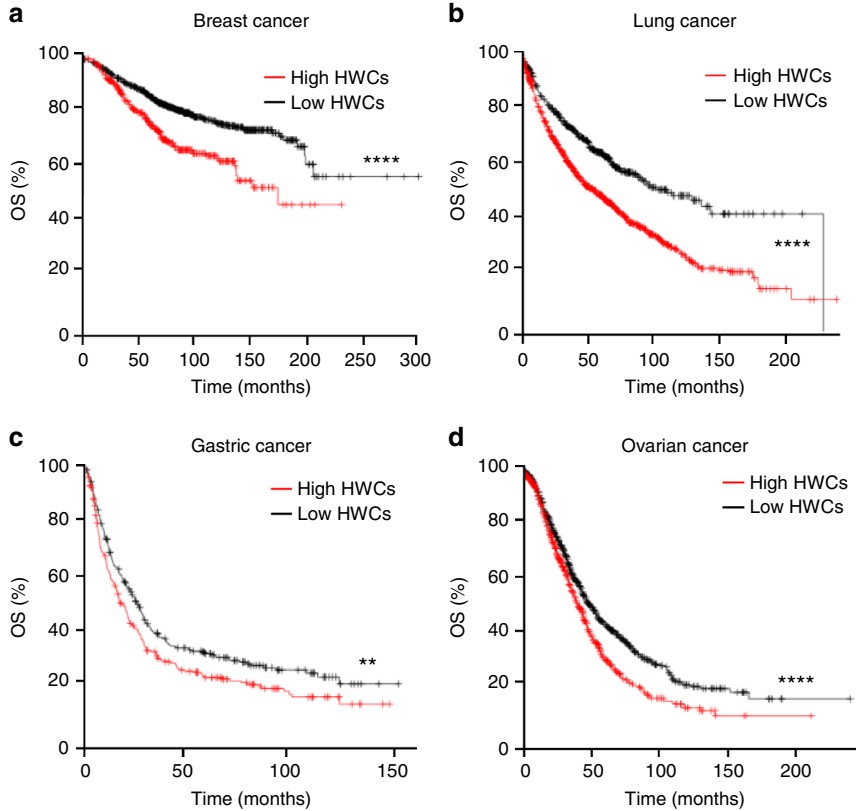

**Fig. 3** The presence of a healing wound-like cytokine response is associated with poor prognosis in human cancers. Kaplan–Meier survival curves showing overall survival (OS) with high (red) and low (black) tumour expression of HWCs (*SPP1*, *IL1B* and *IL6*) for patients with **a** breast cancer ($n = 996$ in HWCs[lo] group and 406 in HWCs[hi] group), **b** lung cancer ($n = 622$ in HWCs[lo] group and 1304 in HWCs[hi] group), **c** gastric cancer ($n = 313$ in HWCs[lo] group and 280 in HWCs[hi] group) and **d** ovarian cancer ($n = 843$ in HWCs[lo] group and 813 in HWCs[hi] group). **P ≤ 0.01, ****P ≤ 0.0001

(Fig. 6k), demonstrating that IL-6 played a key role in the process. IL-6 has been described to augment IL-4R signalling in macrophages[49], which can polarise macrophages to facilitate tumour cell invasiveness[7,50], however pulmonary metastasis was unaffected in 4T1 tumours grown in $Il4r^{-/-}$ mice (Fig. 6l and Supplementary Figure 7d). There were also significantly fewer pulmonary metastases in mice bearing E0771 tumours that were devoid of FAP[+] HO-1[+] TAMs when compared to mice bearing 4T1 tumours (Fig. 6m and Supplementary Figure 7e). These data suggest that HO-1[+] activity by FAP[+] TAMs could play a role in the metastatic spread of tumours, most likely by promoting tumour cell intravasation due to their prevalence in the perivascular region.

**Macrophage HO-1 activity facilitates tumour cell TEM.** To investigate whether the IL-6/HO-1 axis might directly facilitate transendothelial migration of tumour cells, BMDM were exposed to IL-6 to upregulate HO-1 expression (Fig. 4a) and co-cultured with 4T1 tumour cells on an endothelial cell layer in an in vitro transwell assay (Fig. 6n and Supplementary Figure 8a, b). IL-6-exposed macrophages (M(IL-6)) strongly enhanced transendothelial migration of tumour cells, which was HO-1 dependent as SnMP completely abrogated this effect (Fig. 6o). CO, a by-product of HO activity, is a biologically active molecule that can play key roles in cytoprotection through modulating cellular signalling, the mitochondrial electron transport chain, and the generation of reactive oxygen species[51]. To exclude the possibility that SnMP might have affected the viability of the 4T1 cells or macrophages, thereby reducing the number of live migrated cells, 4T1 tumour cells and macrophages were incubated with

increasing doses of SnMP. However, SnMP did not affect the viability of either 4T1 cells or macrophages at any dose or time point tested (Supplementary Figure 8c, d). Also, neither M(IL-6) cells nor SnMP treatment affected the permeability of the endothelial monolayer (Fig. 6p and Supplementary Figure 8e, f), suggesting that HO-1 plays an active role in the migration of the tumour cells through the endothelial cell layer. We assessed which by-product of haem catabolism could account for these effects. Exposure of 4T1 cells to biliverdin or Fe did not increase cellular migration; however, exposure to CO significantly increased the number of 4T1 cells migrating across the endothelial barrier (Fig. 6q), which was also independent of the permeability of the endothelial monolayer (Supplementary Figure 8g). These data suggest that IL-6 stimulated the HO-1-dependent production of CO by macrophages, thereby promoting tumour cell transendothelial migration.

These observations collectively prompt the need to consider evaluating cancer patients for an innate wound healing-like response, for which the cytokine milieu can be informative. This study also provides a potential therapeutic approach using SnMP, a drug that has already been used in the clinic[46,47], to target the ability of FAP[+] HO-1[+] TAMs to facilitate tumour cell intravasation and spread of the disease (Fig. 7).

### Discussion

In this study, we have demonstrated that tumours exploit macrophages into adopting a phenotype similar to that generated by macrophages found in the granulation tissue during the inflammatory response to wounding, which is characterised by

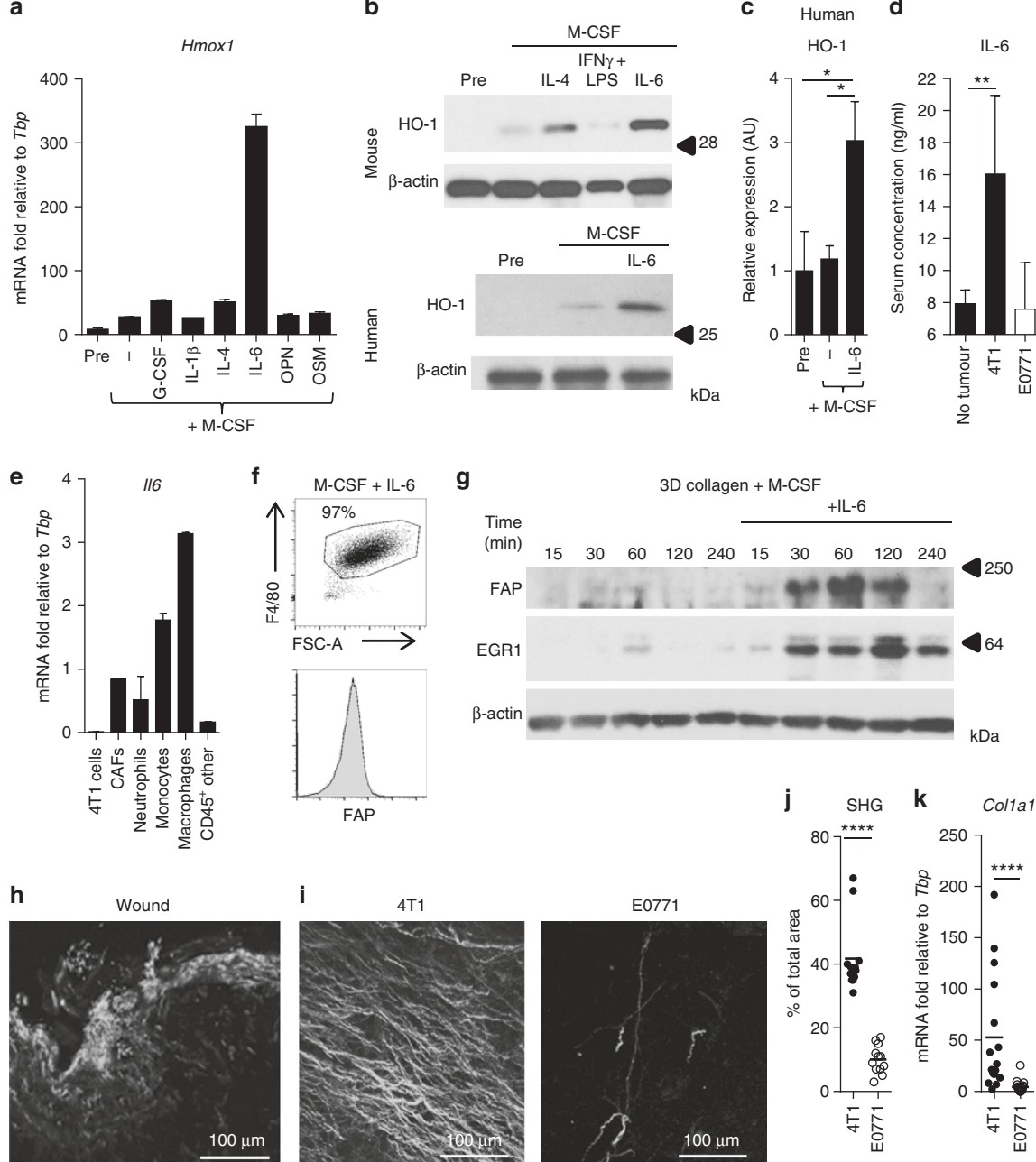

**Fig. 4** IL-6 and collagen drive FAP⁺ HO-1⁺ TAM differentiation in vitro. **a** *Hmox1* mRNA expression relative to the housekeeping gene *Tbp* in bone marrow-derived (BM) cells before (Pre) and after macrophage differentiation in the presence of indicated cytokines at 50 ng/ml for 72 h ($n = 2$ wells; representative of three experiments). **b** Western blot for HO-1 and the loading control β-actin in BM cells before (Pre) and after 72 h culture in the presence of 50 ng/ml M-CSF with/without 50 ng/ml IL-4, IFNγ + LPS or IL-6 (top) and human peripheral blood (PB)-derived macrophages before (Pre) and after 72 h culture in the presence of 50 ng/ml M-CSF with/without 50 ng/ml IL-6 (bottom). **c** Quantitation of HO-1 expression in human PB-derived macrophages from individual healthy volunteers ($n = 5$) as described in **b**. **d** Serum IL-6 concentrations in tumour-free mice and in those bearing late stage (>20 days post inoculation) 4T1 (black) or E0771 (white) tumours ($n = 5$) as determined by ELISA. **e** Representative *Il6* mRNA expression relative to *Tbp* in FACS-sorted tumoural cell populations from a 4T1 tumour ($n = 2$ wells, representative of duplicate experiments). **f** Representative gating, purity and FAP surface expression of BM cells cultured for 72 h in the presence of 50 ng/ml M-CSF and IL-6. **g** Representative western blot analysis for FAP (top), EGR1 (middle) and β-actin (bottom) in BM cells cultured for 72 h with 50 ng/ml M-CSF with/without 50 ng/ml IL-6, and subsequently incubated for indicated times on 3D murine type I collagen. **h, i** Representative images of second-harmonic generation (SHG) imaging of fibrillar collagen in frozen sections of wounded skin (**h**) and of 4T1 and E0771 tumours (**i**). **j** Quantitation of SHG signal in 4T1 and E0771 tumours ($n = 12$ tumours (>20 days post inoculation) in each group; average across multiple fields of view). **k** *Col1a1* mRNA expression relative to *Tbp* in 4T1 ($n = 16$) and E0771 ($n = 18$) tumours (>20 days post inoculation). Bar charts are presented as mean + s.d. *$P \leq 0.05$, **$P \leq 0.01$, ****$P \leq 0.0001$

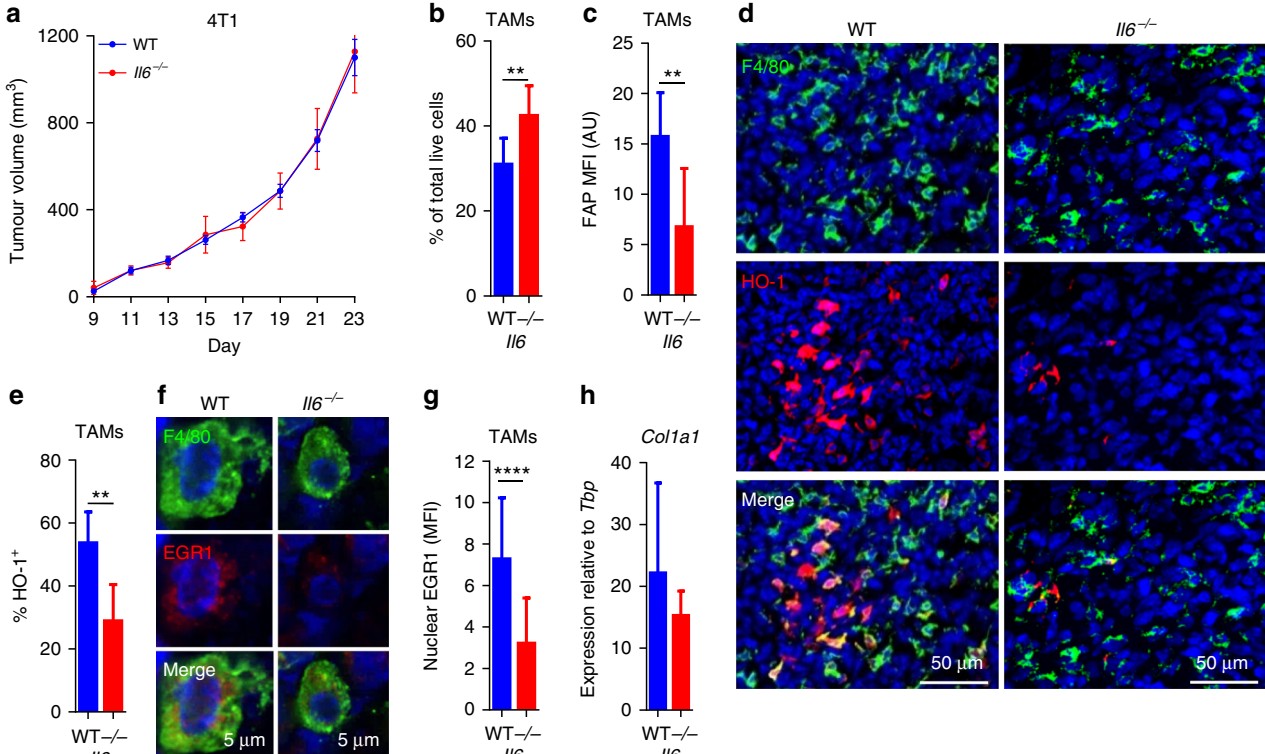

**Fig. 5** IL-6 regulates the FAP+ HO-1+ TAM phenotype in vivo. **a** Example growth curves of 4T1 tumours grown in WT (blue) or $Il6^{-/-}$ (red) mice ($n = 8$ per condition). At day 23 post inoculation of 4T1 cells, tumours were excised and analysed. **b**, **c** The abundance of live CD45+ F4/80+ TAMs (**b**) and their median fluorescence intensity (MFI) of FAP surface expression with FMO background staining subtracted (**c**) ($n = 11$ WT mice and 7 $Il6^{-/-}$ mice). **d** Representative frozen sections of 4T1 tumours stained with DAPI (nuclei; blue) and antibodies against F4/80 (green) and HO-1 (red). **e** Quantification of the fraction of F4/80+ cells co-expressing HO-1 ($n = 5$ tumours) as shown in **d**. **f** Representative frozen sections stained with DAPI (nuclei; blue) and antibodies against F4/80 (green) and EGR1 (red). **g** Quantification of nuclear EGR1 in F4/80+ cells ($n = 20$ areas across three tumours per condition). **h** $Col1a1$ mRNA expression in 4T1 tumours relative to the housekeeping gene $Tbp$. Growth curve in **a** is presented as mean ± s.e.m. and bar charts represent mean + s.d. **$P \leq 0.01$, ****$P \leq 0.0001$

co-expression of FAP and HO-1, which in the context of cancer can facilitate transendothelial migration of tumours cells.

Harold Dvorak's seminal observation that tumours and healing wounds display inherent similarity[18], is now a concept which is deeply embedded in our understanding of the stromal response in cancer[52]. Gene signatures associated with 'healing wounds' have previously been identified in breast cancer and have been linked to poor prognosis[53,54]. In the present study, we specifically describe cytokine genes that are associated with the inflammatory phase of a wound response, comprising $Il1B$, $IL6$ and $SPP1$, which are upregulated in the inflammatory phase of both human and mouse wounds. Expression levels of these three genes were sufficient to segregate patients with poor prognosis for breast, lung, gastric and ovarian cancer, highlighting the prognostic significance in cancer progression.

Very little is known about the FAP+ HO-1+ subset of macrophages, which can be found in both human and murine breast tumours. Here, we identified their appearance as a wound healing response phenotype associated with the granulation tissue, which provides significant biological insight and context to understand their origin and function. FAP+ HO-1+ macrophages first appear within the granulation in the initial days after wounding, during the acute inflammatory phase of the response, however their presence is maintained during the wound healing response. In the current study we investigated the dorsal wound; however, the cytokine response can also be dictated, and fine-tuned, by the anatomical location of the wound site, such as was observed between the skin

and mucosal surface[28]. The presence of FAP+ HO-1+ macrophages at other wound sites, and their specific role within the granulation tissue, although not covered by the current study, remain important questions to consider.

We have demonstrated IL-6 to be a key regulator of the FAP+ HO-1+ macrophage phenotype, and this cytokine is also expressed directly by the macrophages in 4T1 tumours. IL-6 has the potential to regulate HO-1 expression through both autocrine and paracrine signalling, concurrently licensing FAP expression in macrophages in a collagen-rich microenvironment. FAP is a dipeptidyl peptidase capable of degrading type I collagen[11,12], which is coherent with our observation that FAP expression is induced in part by exposure to type I collagen, its substrate. Type I collagen is the major collagen of the dermal layer of the skin[40], and its expression in cancer has been associated with poor prognosis in patients[55]. FAP has long been associated with tumour progression and wound healing in animal models[56,57], and is also associated with poor clinical outcome in patients with cancer[58,59]. Although in the current study, FAP has only been used as a marker, its expression could feasibly facilitate a macrophage's ability to migrate through the collagen networks found in the dermis and in the tumour microenvironment, similar to that demonstrated for FAP-expressing fibroblasts[60]. FAP expression on TAMs could serve as a marker of a regenerative-like tumour microenvironment.

TAMs have been demonstrated to play a critical role in metastatic spread[7,61], and tumour cells and TAMs are closely associated during intravasation[43]. A well-characterised example of this

interaction is through a paracrine loop involving M-CSF released from tumour cells, and epidermal growth factor (EGF) released by TAMs[62]. The importance of the M-CSF/EGF paracrine loop in metastasis has been clearly demonstrated by mammary xenograft models using a conditional knockout of CSF-1[62]. We demonstrate that HO-1 expression, which is upregulated in response to the HWC IL-6, plays a role in facilitating transendothelial migration of

tumour cells, thereby supporting their metastatic spread. HO-1 expression also correlates with metastasis in human cancer[63] and has previously been implicated with metastasis in murine models[64]. While our results indicate that FAP⁺ HO-1⁺ TAMs facilitate tumour cell intravasation, supported by their presence in the perivascular region of the tumour, others have also described a role for lung-resident HO-1⁺ macrophages in facilitating the

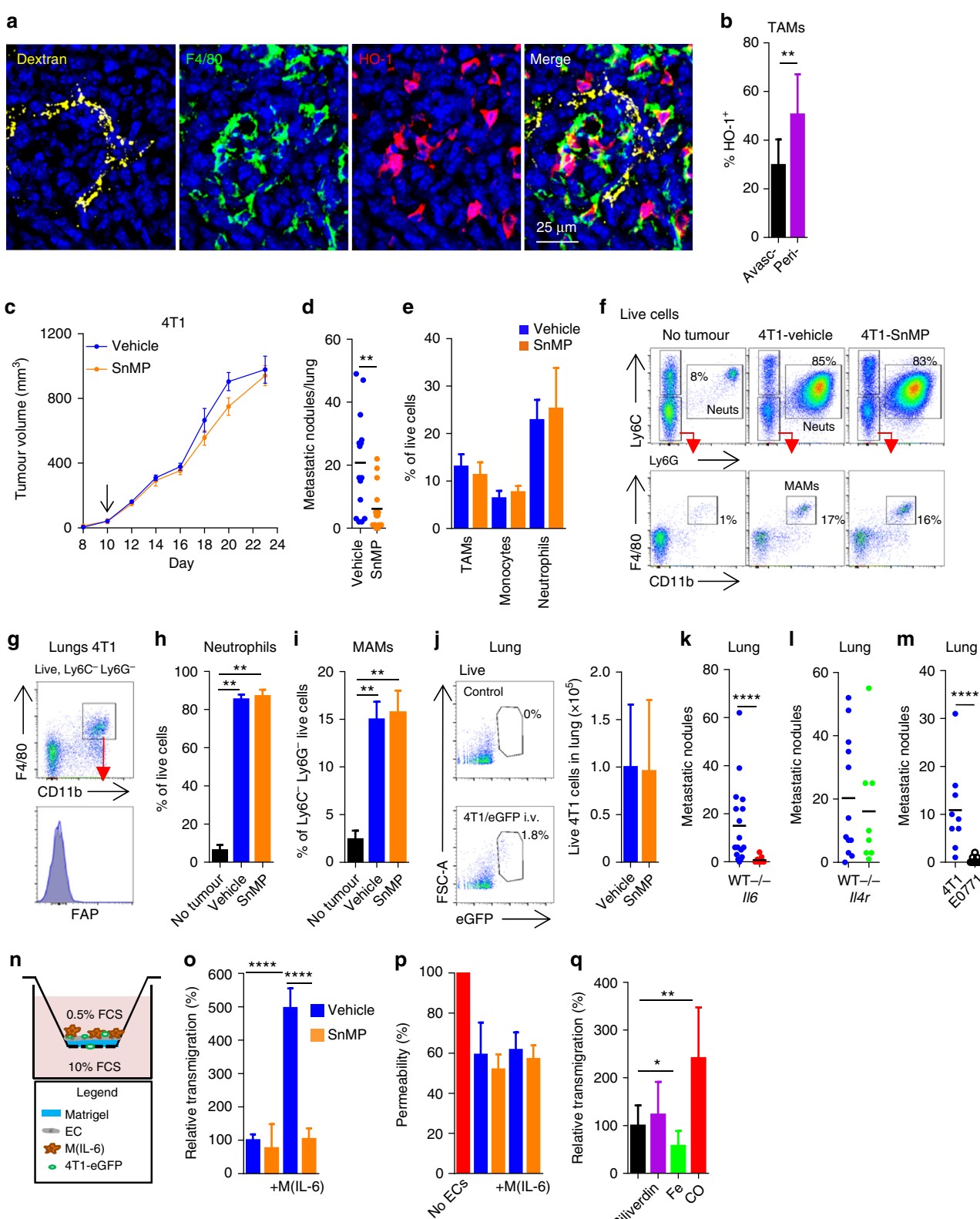

extravasation of metastatic tumour cells entering the lung[64]. Despite our results suggesting that lung colonisation of 4T1 cells was HO-independent, these findings highlight the broader physiological role of HO activity in transendothelial migration, which is exploited by different tumours.

We have demonstrated that the haem catabolite CO, which is capable of modulating cellular signalling and functions[64–68], was capable of facilitating transendothelial migration of 4T1 cells, which is consistent with previous reports[64]. The mechanism through which this was achieved was independent of the role of HO-1 in vasorelaxation[69] as HO-1-expressing macrophages had no observable effect on the permeability of an endothelial layer. HO-1 has also been demonstrated to play an active role in injury repair[70], to facilitate wound closure[71] and to improve the migratory potential of keratinocytes[71], which may explain their ability to facilitate transendothelial migration of 4T1 cells.

In 4T1 tumours, the TAMs were the major, but not the only stromal source of IL-6. Interestingly, macrophages have been demonstrated to secrete IL-6 when co-cultured with 4T1 cells[72], suggesting that TAMs can be coerced into adopting this response by a direct interaction. There are characterised interactions which can result in IL-6 expression by macrophages[73,30], although the specific interaction was not investigated in this current study, it is an important question to resolve. However, this study prompts the wider need to consider the importance of IL-6 in the macrophage response in both wound healing and cancer.

In the present study, we have demonstrated that FAP+ HO-1+ TAMs found in the microenvironment of human and murine breast cancers represent an innate wound healing response phenotype. Their phenotype is analogous to the response of macrophages found in the granulation tissue after wounding. FAP+ HO-1+ TAMs have previously been demonstrated to play a fundamental role in immune suppression in the tumour microenvironment[3], a role also attributed to the expression of HO-1 in the wound[74]. Furthermore, we demonstrate that HO-1 expression by these macrophages can facilitate metastatic spread. These data collectively suggest that FAP+ HO-1+ macrophages reflect a potent pro-tumorigenic TAM phenotype. However, the full diversity of cancers and other diseases in which FAP+ HO-1+ macrophages can play a role has yet to be established. This study has started to shed light on the contextual origin of FAP+ HO-1+ TAMs and their exploitation by the tumour to facilitate metastatic spread of the disease. Moreover, we demonstrate that pharmacological inhibition of HO-1 is a viable therapeutic strategy to prevent their role in metastasis.

## Methods

**Mice**. Balb/c mice WT and homozygous null (−/−) for *Il6* or *Il4r* were obtained from Charles River. Balb/c *Rag2*−/− were a gift from Professor Adrian Hayday (KCL). All mice used for experiments were female and randomly assigned to non-blinded treatment groups. Cohort sizes were informed by prior studies[3,15]. Superficial wounding of the skin was performed using an 8 mm punch biopsy needle (Stiefel Instruments) under analgesia and general anaesthesia. Experiments were performed at least in duplicate;. The use of animals was approved by the Ethical Review Committee at King's College London and the Home Office, UK.

**Cell lines**. 4T1 mammary adenocarcinoma cells[6] and 3B-11 endothelial cells were obtained from ATCC. E0771 mammary adenocarcinoma cells[20,21] were a gift from Prof. Anne Ridley (King's College London). Cell lines were confirmed to be mycoplasma free using the MycoAlert Mycoplasma Detection Kit (Lonza) and were cultured in RPMI 1640 (Gibco) supplemented with 10% FCS.

**Transduction of cell lines**. The pMIG retroviral vector was a gift from Prof. Douglas Fearon, Cold Spring Harbor Laboratory, New York, and was modified to contain Click Beetle luciferase and enhanced green fluorescent protein (eGFP) separated by a viral P2A sequence. Click Beetle luciferase cDNA was cloned from the vector pCBG99 (Promega) using forward primer (P1) 5′-GTGA AGCGTGAGAAAAATGTCA-3′ and reverse primer (P2) 5′-TGAAGTTAGTA GCTCCGCTTCCACCGCCGGCCTTCTCCAA-3′, which also incorporated a P2A sequence overhang. Click Beetle luciferase cDNA was joined to P2A using overlap PCR with a synthetically derived P2A cDNA sequence (Integrated DNA Technologies) using P1 and reverse primer (P3) 5′-CTCCTCGCCCTTGCTCACA GGTCCAGGGTTCTCCTCC-3′, which also incorporated an eGFP sequence overhang. eGFP cDNA was cloned from the vector pEGFP-N1 using forward primer (P4) 5′-GTGAGCAAGGGCGAGGAG-3′ and reverse primer (P5) 5′-C TTGTACAGCTCGTCCATGC-3′. Click Beetle luciferase/P2A and eGFP cDNA were joined using PCR with primers P1 and P5. Finally, *Xho*1 and *Hin*dIII restriction enzyme sites were added to the reporter construct cDNA using PCR with the respective forward 5′-CTCGAGATGGTGAAGCGTGAGAAAA ATGTCA-3′ and reverse 5′-AAGCTTTTACTTGTACAGCTCGTCCATGC-3′ primers. The PCR product was then subcloned using Zero Blunt™ Topo™ Kit (Thermo Fisher Scientific), and subsequently inserted into pMIG using the restriction enzymes *Xho*I and *Hin*dIII (New England Biolabs) and T4 DNA ligase (Thermo Fisher Scientific). Full-length murine *Csf3*, the gene for G-CSF, was cloned using PCR with forward primer 5′-TTAGCGATCAAGATCTACCATG GCTCAACTTTCTGCCCAGAG-3′ incorporating a *Bgl*II site, and reverse primer 5′-CGTGAATCGACTCGAGCTAGGCCAAGTGGTGCAGAG-3′ incorporating a *Xho*I site, from purified 4T1 tumour cell mRNA which had been converted to cDNA using the RT² first strand kit (Qiagen). *Csf3* cDNA was subsequently inserted into pMIG using the restriction enzymes *Bgl*II and *Xho*I (New England Biolabs) and T4 DNA ligase (Thermo Fisher Scientific). These restriction enzyme sites maintained an endogenous Thy1.1 reporter element within the pMIG vector. Empty vector containing Thy1.1 alone was used as a control. Retroviral particles were produced, and 4T1 and E0771 cells were transduced as previously described[15]. eGFP+ 4T1 and Thy1.1+ and Thy1.1/G-CSF+ E0771 cells were selected by flow cytometry for their expression of eGFP or Thy1.1, respectively, and subsequently expanded in culture. G-CSF was confirmed to be secreted using the mouse G-CSF ELISA set (Insight) according to the manufacturers' protocol on conditioned media from the cell lines.

**Fig. 6** Perivascular FAP+ HO-1+ macrophages can facilitate transendothelial migration of tumour cells through their HO activity. **a** Representative images of frozen sections of a 4T1 tumour stained with DAPI (nuclei; blue) and antibodies against F4/80 (green) and HO-1 (red); functional vasculature was labelled in vivo using dextran-FITC (yellow). **b** Quantification of F4/80+ TAMs co-expressing HO-1 in avascular and perivascular regions (*n* = 12 sections across four tumours in each group). **c** Tumour growth curves of mice bearing established 4T1 tumours treated with vehicle (blue) or 25 μmol/kg/day SnMP (orange) (*n* = 6), arrow marks the initiation of treatment. **d** Metastatic nodules on the lung surface of mice bearing 4T1 tumours treated with vehicle (*n* = 16) or 25 μmol/kg/day SnMP (*n* = 15) at day 24 post inoculation of tumour cells (data from three independent experiments presented). **e–i** 4T1 tumours or lungs in mice treated with vehicle (blue) or SnMP (orange) as described in **c** were analysed. **e** Abundance of TAMs, monocytes and neutrophils in enzyme-dispersed tumours (*n* = 5 per condition). **f** Representative flow cytometry gating strategy for MAMs. **g** Representative surface FAP staining (blue) and isotype control staining (grey shaded) of MAMs in the lungs of 4T1 tumour-bearing mice (day 24). **h, i** Abundance of lung neutrophils (**h**) and MAMs (**i**) (*n* = 4 per condition). **j** Representative flow cytometry gating strategy (left) and abundance (right) of 4T1-eGFP cells in enzyme-digested lungs 48 h after i.v. injection of 4T1-eGFP in mice treated with vehicle (blue) or 25 μmol/kg/day SnMP (orange) (*n* = 10 mice). **k–m** Abundance of metastatic nodules on the surface of lungs in mice bearing 4T1 tumours grown in WT (*n* = 17) and *Il6*−/− (*n* = 9) mice (**k**), WT (*n* = 12) and *Il4r*−/− (*n* = 8) mice (**l**), or in syngeneic mice bearing 4T1 (*n* = 9) or E0771 (*n* = 9) tumours (**m**). **n** Schematic representation of the transendothelial migration assay; EC endothelial cells. **o, p** Relative transendothelial migration of 4T1-eGFP cells (**o**) and permeability of the EC layer to albumin (**p**) in the presence or absence of HO-1+ Ly6Chi monocyte-derived macrophages (M(IL-6)) with/without 25 μM SnMP (orange) (*n* = 4 wells; representative of duplicate experiments) over a 24 h period. **q** Relative transendothelial migration of 4T1-eGFP cells in the absence (*n* = 12) or presence of 20 μM Iron (II) chloride (*n* = 8), 5 μM biliverdin (*n* = 8) or 250 ppm CO (*n* = 3) (pooled data from separate experiments). Bar charts represent mean + s.d. *P < 0.05, ** P < 0.01, ****P < 0.0001

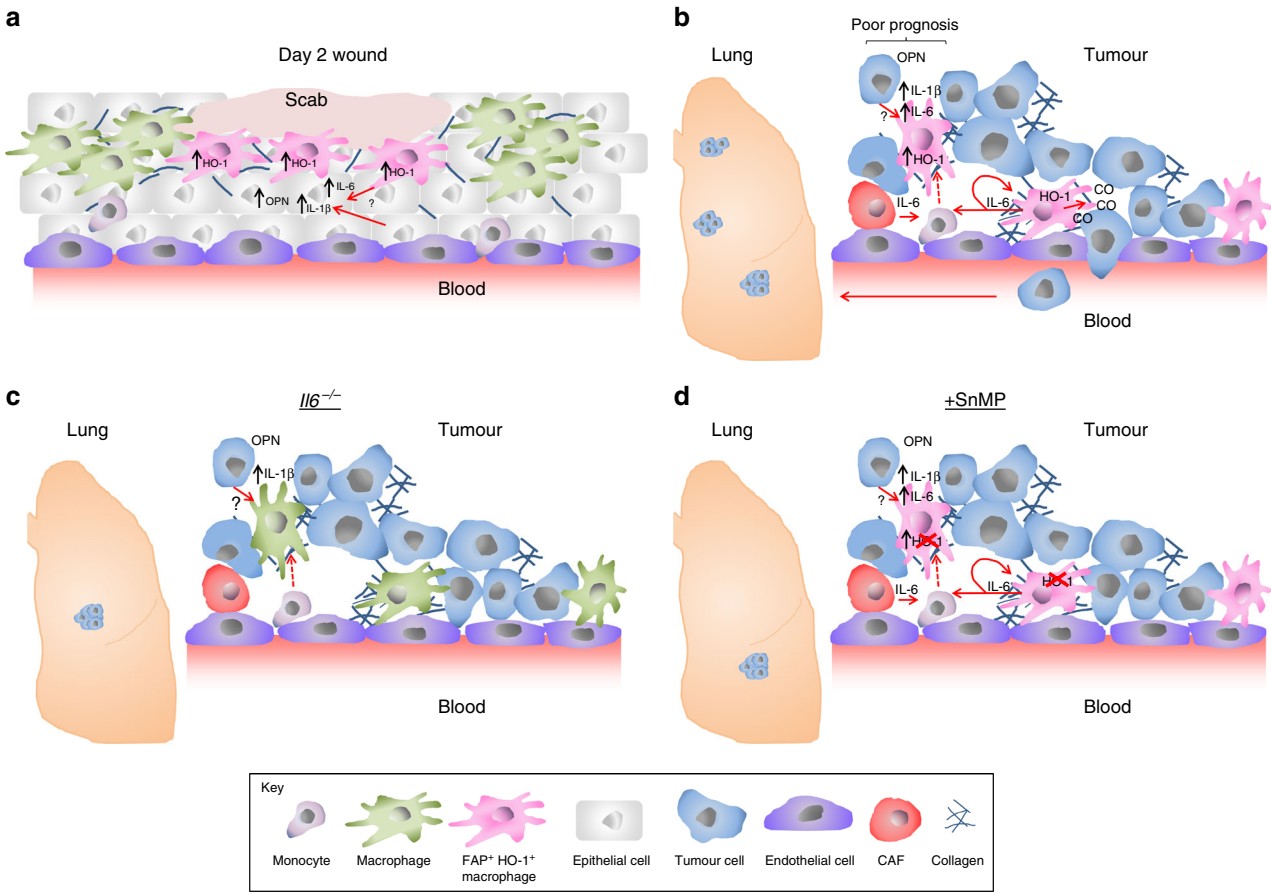

**Fig. 7** Macrophages are exploited from an innate wounding response to facilitate metastasis. **a** The acute wounding inflammatory response results in a microenvironment with high levels of osteopontin (OPN), IL-1β and IL-6. In response to IL-6, in combination with the collagen-rich environment of the skin, the macrophages adopt an FAP⁺ HO-1⁺ wound response phenotype. **b** Tumours exploit the acute wound healing response of macrophages, by inducing IL-6 expression in these cells, as well as other stromal cells, by a currently unknown mechanism. IL-6 signals to monocytes/macrophages resulting in upregulation of HO-1 expression. The tumour microenvironment is also rich in collagen, which licences FAP expression on TAMs in an IL-6-dependent manner. FAP⁺ HO-1⁺ TAMs are predominantly located in the perivascular region and facilitate transendothelial migration of tumour cells through their production of CO, facilitating tumour cell intravasation into the bloodstream and eventual colonisation in the lung. **c** In the absence of IL-6 (*Il6*⁻/⁻ mice), the tumour is compromised in its ability to generate the wound healing response TAM phenotype, and lung metastasis is reduced. **d** FAP⁺ HO-1⁺ TAMs can be therapeutically targeted using SnMP to inhibit their production of CO, which prevents transendothelial migration of the tumour cells and subsequent lung colonisation

**Tumour studies**. 4T1 or E0771 cells ($2.5 \times 10^5$ in 100 μl RPMI) were orthotopically implanted by subcutaneous injection into the mammary fat pad of female mice that were 6–8 weeks of age. The sizes of the subsequent tumours were determined as previously described[15]. Sn (IV) mesoporphyrin IX dichloride (SnMP; Frontier Scientific) was prepared freshly on the day of injection as previously described[3] and administered daily at 25 μmol/kg by intraperitoneal injection. Blood samples were taken from mice in EDTA-coated Microvette™ tubes (Sarstedt). Plasma was extracted from blood samples by centrifugation at $2000 \times g$ for 5 min and IL-6 quantitated using the mouse IL-6 ELISA set (Insight Biotechnology) according to the manufacturer's protocol. Tumour tissue and other organs were enzyme-digested to release single cells as previously described[15]. Spontaneous lung meta-static surface nodules were quantified by manual counting in a blinded manner.

**In vivo metastasis assay**. Wild-type Balb/c mice that were 6–8 weeks of age were treated with 25 μmol/kg/day SnMP or vehicle control from 72 h prior to tumour cell challenge. On day zero, $5 \times 10^5$ 4T1-eGFP cells in 200 μl Dulbecco's PBS (Thermo Fisher Scientific) were injected intravenously via the tail vein. After 48 h, mice were killed and the lungs were dissociated by enzyme dispersion to release a single-cell suspension as previously described[15]. 4T1-eGFP cells that had colonised the lung were identified by their eGFP expression and quantified using counting beads by flow cytometry analysis.

**Magnetic-bead isolation of macrophages and monocytes**. Single cells released from murine tumour tissue using enzyme dispersion, or crushed from a spleen of a 4T1 tumour-bearing mouse, were pelleted by centrifugation at $500 \times g$ and red blood cells

(RBC) were lysed using RBC lysis buffer (eBioscience) according to the manufacturer's protocol. Human peripheral blood mononuclear cells (PBMCs) were obtained from anonymised human buffy coats as supplied by the NHS Blood and Transplant (London, UK). Buffy coats were diluted 1:2 with calcium and magnesium-free PBS (Gibco), then the PBMC fraction was isolated using a Lymphoprep (Axis-Shield) density gradient spun at $800 \times g$ for 30 min at RT. Cells were pelleted and resuspended in PBS containing 2 mM EDTA, 1% FCS. All subsequent steps were performed on ice for 30 min unless otherwise stated. Murine Fc receptors were blocked using 5 μg/ml anti-CD16/32 (2.4G2, Tonbo Biosciences) prior to staining with F4/80 PE (BM8; to isolate macrophages) or Ly6C PE (HK1.4; to isolate monocytes) at 1 μg/ml, followed by anti-PE MicroBeads (Miltenyi Biotec). CD14⁺ human monocytes were incubated directly with anti-human CD14 MicroBeads (Miltenyi Biotec). Both human and murine cells were subsequently isolated using a MidiMacs separator and LS columns (Miltenyi Biotec) according to the manufacturer's protocol.

**In vitro-derived macrophages**. Murine bone marrow (BM) was flushed from the femur and tibia using a syringe and needle, and RBCs were lysed using RBC lysis buffer (eBioscience). Complete murine BM, or splenic murine Ly6C⁺ cells or PBMC-derived human CD14⁺ MACS-sorted monocytes were plated in RPMI 10% FCS, 1× penicillin/streptomycin (Sigma-Aldrich), 50 ng/ml recombinant murine (Bio-techne) or human (Peprotech) M-CSF at $1 \times 10^6$ cells/well on 6-well plates for subsequent mRNA and protein analyses. Where viable macrophages were required for ongoing experiments, cells were plated at $5.5 \times 10^6$ cells on 6 cm non-tissue culture-treated plates. Additional murine cytokines, IL-4 (Bio-techne) and IFN-γ (Bio-techne), human IL-6 (Peprotech), or LPS (Sigma-Aldrich) were added where indicated in the figure legends at 50 ng/ml unless

stated otherwise. After 72 h in culture, macrophage purity was assessed by flow cytometry. Macrophages differentiated in the presence of M-CSF only are referred to as M0 cells, and macrophages differentiated in the presence of M-CSF and IL-6 as M(IL-6) cells.

**Collagen matrix assembly.** A collagen matrix was prepared by mixing 2 mg/ml type I collagen (BIO RAD) in 50 mM acetic acid with 1.5 parts ice cold RPMI, 10% FCS, 33.3 mM NaOH, and 200 µl was transferred to each well of a 24-well plate for polymerisation for 30 min at 37 °C. Differentiated macrophages (described above) were re-plated onto the collagen matrix in RPMI, 10% FCS, 50 ng/ml recombinant murine M-CSF (Bio-techne) at $2 \times 10^5$ cells/well.

**Transendothelial migration assays.** Transendothelial migration assays were performed as described previously[8] with modifications. Briefly, transwell inserts with 8 µm pore size (Corning) were coated with 125 µg/ml Matrigel (Corning). An endothelial barrier was established by culturing $2 \times 10^4$ 3B-11 endothelial cells in RPMI with 10% FCS on the Matrigel-coated inserts. After 72 h in culture, medium was replaced with RPMI, 0.5% FCS containing $2 \times 10^4$ 4T1-eGFP cells, and RPMI, 10% FCS was added to the bottom chamber. Where indicated, $2 \times 10^4$ splenic-derived M(IL-6) macrophages were added to the upper chamber and allowed to adhere for 2 h in RPMI, 10% FCS. The media was removed and $2 \times 10^4$ 4T1-eGFP cells were added in RPMI, 0.5% FCS. These experiments were performed in the presence or absence of 50 ng/ml IL-6 and M-CSF, 25 µM SnMP, 20 µM iron (II) chloride (Sigma-Aldrich), 5 µM biliverdin hydrochloride (Sigma-Aldrich), or a gas mix of 250 ppm CO, 5% $CO_2$, $N_2$ balance (BOC) in a hypoxia incubator chamber (Stemcell Technologies). After 72 h of incubation at 37 °C, cells were removed from the upper chamber using a cotton swab. Transmigrated cells were dissociated from the bottom of the transwell using enzyme-free dissociation buffer (Thermo Fisher Scientific). Collected cells were mixed with AccuCheck counting beads (Thermo Fisher Scientific) and 7-amino actinomycin D (7AAD) (Sigma-Aldrich) prior to analysis by flow cytometry. The permeability assay was performed as described previously[8]. In brief, permeability was measured using Evans Blue-conjugated bovine serum albumin (BSA) (4%) in PBS, placed into the upper chamber of a Matrigel-coated transwell insert with 0.4 µm pore size (Corning), and phenol red-free RPMI, 10% FCS was added to the bottom chamber. Presence of Evans Blue-BSA in the bottom chamber was assessed at indicated time points by absorbance at 620 nm on a NanoDrop™ 2000 spectrophotometer (Thermo Fisher Scientific).

**Cell viability MTT assays.** Bone marrow-derived macrophages (BMDM) were plated at a density of $1 \times 10^6$ cells per well of a 6-well plate in RPMI 1640 (Thermo Fisher) supplemented with 10% FCS, 10 ng/µl M-CSF (Bio-techne) and penicillin/streptomycin (Sigma-Aldrich); 4T1 cells were plated at a density of $3.5 \times 10^5$ (24 h assay) or $1 \times 10^3$ (4 day assay) in 24-well tissue culture plates in RPMI, 10% FCS. Cells were given 24 h to attach, and were then incubated with media containing either SnMP or vehicle. At the indicated time point, media was exchanged with phenol-free RPMI (Sigma-Aldrich) supplemented with 10% FCS and 0.5 mg/ml MTT (3-(4,5-dimethylthiazol-2-yl)-2,5-diphenyltetrazolium bromide) reagent and incubated for 1 h. MTT-RPMI media was aspirated and the formazan crystals (generated from metabolised MTT) were solubilised with dimethyl sulfoxide and the absorbance measured at 538 nm on a Fusion alpha-FP spectrophotometer (Perkin-Elmer) or NanoDrop™ 2000 spectrophotometer (Thermo Fisher Scientific). All values were normalised to the vehicle control-treated wells.

**Flow cytometry and cell sorting.** Flow cytometry was performed as previously described[15]. The following antibodies were purchased from eBioscience and were used at 1 µg/ml unless stated otherwise: CCR2 APC (475301; 1:10; Bio-techne), CD11b Brilliant Violet 510™ (M1/70; Biolegend®), CD11b eFluor®450 (M1/70), CD11c APC (N418), CD14 APC (Sa2-7), CD45 APC-eFluor™780 (30-F11), F4/80 APC/PE/FITC (BM8), IL-4R PE (1:10; Bio-techne), Ly6G FITC (1A8; Biolegend®), MHCII APC (M5/114.15.2), MMR APC (HK1.4), Ly6C PE (HK1.4), Thy1.1 eFluor 450 (HIS5a), Thy1.2 eFluor 450 (53-2.1). Where stated, the following corresponding isotype control antibodies at equivalent concentrations to that of the test stain were used within fluorescence minus one staining panels: goat IgG APC and PE (Bio-techne), rat IgG2b APC and eFluor® 450 (eB149/10H5), rat IgG2a APC and PE (eBR2a) and Armenian Hamster IgG APC (eBio299Arm). FAP was stained as previously described[15]. Dead cells and red blood cells were excluded using 1 µg/ml 7AAD (Sigma-Aldrich) and anti-Ter-119 PerCP-Cy5.5 (Ter-119), respectively. Flow cytometry was performed on a BD FACS Canto II (BD Biosciences) or cells were sorted for subsequent analyses using a BD FACSAria (BD Biosciences). Data were analysed using FlowJo software (Freestar Inc.). Cell populations were distinguished based upon the following surface characteristics: CD45+ (immune cells), CD11b+ F4/80hi (macrophages), CD11b+ Ly6G− Ly6C+ (monocytes), CD11b+ Ly6G+ (neutrophils), CD3ε+ (T cells), CD3ε+ CD4+ (CD4+ T cells), CD3ε+ CD8α+ (CD8+ T cells), CD19+ (B cells), CD45− Thy1+ (CAFs), CD45− CD31+ (endothelial cells), CD45− Thy1− CD31− (tumour cells) and CD45+ CD11b+ Ly6C−Ly6G− F4/80hi (MAMs).

**Western blot.** Cells were lysed and SDS-PAGE/western blots were conducted as previously described[6] with the following primary antibodies at 1:1000 unless stated

otherwise: HO-1 (mouse: EP1391Y, Origene; human: 41211, Bio-techne), FAP (AF3715, Bio-techne), EGR1 (15F7, Cell Signalling Technology), β-actin, 1:5000 (Abcam) and detected using the following horseradish peroxidase-conjugated secondary antibodies: goat anti-rabbit IgG (H+L), 1:25,000 (Thermo Fisher Scientific), anti-sheep IgG, 1:2500 (Bio-Techne). Protein bands were detected using Luminata™ Crescendo Western HRP substrate (Millipore) and CL-XPosure™ Film (Thermo Fisher Scientific). Densitometry was performed using ImageJ software. The uncropped western images used in this paper can be seen in Supplementary Figure 4.

**Quantitative real-time PCR.** mRNA was extracted and quantitative real-time PCR was performed as previously described[3] using the following primers/probes purchased from Thermo Fisher Scientific: *Csf3 Mm00438335_g1*, *Col1a1 Mm00801666_g1*, *Col3a1 Mm01254476_m1*, *Csf3 Mm00438335_m1*, *Fn1 Mm01256744_m1*, *Hmox1 Mm00516005_m1*, *Il1b Mm00434228_m1*, *Il4 Mm00445259_m1*, *Il6 Mm00446190_m1*, *Il10 Mm01288386_m1*, *Il13 Mm00434204_m1*, *Il24 Mm00474102_m1*, *Osm Mm01193966_m1*, *Spp1 Mm00436767_m1*, *Tbp Mm01277045_m1*, *Tnc Mm00495662_m1*. Expression is represented relative to the housekeeping gene Tata-binding protein (*Tbp*). An extracellular matrix and adhesion molecule RT² profiler PCR array (PAMM-013Z; Qiagen) was also utilised, for this mRNA from 4T1 and E0771 tumour tissue was converted to cDNA using RT² first strand kit (Qiagen), and assays run using RT² SYBR Green ROX qPCR Mastermix (Qiagen) according to the manufacturer's protocol. Gene expression was measured using an ABI 7900HT Fast Real Time PCR instrument (Thermo Fisher Scientific).

**Illumina microarray and bioinformatics analysis.** Purified mRNA for the respective macrophage populations was isolated using the PureLink® RNA Mini Kit (Ambion) according to the manufacturer's protocol. The purity of the isolated mRNA was assessed using a NanoDrop™ 2000 spectrophotometer (Thermo Fisher Scientific) and the quality and integrity using an Agilent 2100 Bioanalyzer (Agilent Technologies). mRNA was converted to cDNA, then subsequently amplified using the Ovation® PicoSL WTA system V2 (NuGen), biotinylated using the Encore® BiotinIL Module (NuGen) and then hybridised to MouseWG-6 V2.0 Beadchip microarray (Illumina). Following hybridisation, the arrays were washed, blocked and stained with streptavidin-Cy3 using the whole-genome gene expression direct hybridisation assay (Illumina). Microarrays were run on an Illumina iScan system, raw fluorescence signals were collected using GenomeStudio (Illumina), and the data imported into Partek Genomics Suite for analysis. Background was subtracted from the raw data and fluorescence signals were normalised using the quantiles method[75].

**Immunofluorescence.** Sections of fresh frozen human breast carcinoma (all identified as grade 3 invasive ductal carcinoma, two basal-like and one HER2+), mouse mammary tumours or wounded skin embedded in OCT were fixed in 4% paraformaldehyde in PBS (Gibco) for 10 min at RT. Immunofluorescence was performed as previously described[15] with the inclusion of 0.2% Triton X-100 in the blocking buffer. The following antibodies were used at 1:100 dilutions unless stated otherwise: F4/80 (C1:A3-1, Bio-RAD), HO-1 (EP1391Y, Origene) or HO-1 (10701-I-AP; Proteintech Group), FAP (AF3715, Bio-techne), EGR1, 1:50 (15F7, Cell Signalling Technology), CD14 (61D3, eBioscience), CD11b (ICRF44, eBioscience). Primary antibodies were detected using Cy™3 donkey anti-sheep IgG, 1:100 (Jackson ImmunoResearch, reconstituted at 1 mg/ml), or the following donkey IgG antibodies purchased from Thermo Fisher Scientific and used at 1:200: AlexaFluor® 488 anti-rabbit IgG, AlexaFluor® 488 anti-rat IgG, AlexaFluor® 647 anti-mouse IgG, AlexaFluor® 647 anti-rabbit IgG. Viable blood vessels were visualised in mice through i.v. injection of FITC-conjugated dextran (MW20,000, Sigma-Aldrich) 30 min prior to sacrifice. Nuclei were stained using 1.25 µg/ml 4′,6-diamidino-2-phenylindole, dihydrochloride (DAPI) (Thermo Fisher Scientific). Images were acquired using a Nikon Eclipse Ti-E Inverted spinning disk confocal microscope system and associated NIS Elements software, and total, nuclear and cytoplasmic staining were quantified using ImageJ. Perivascular TAMs were analysed as previously described[9] with modifications. F4/80+ cells present within a 75 µm radius of a given blood vessel were defined as 'perivascular', and F4/80+ cells present in areas beyond this region were defined as 'avascular'. Co-localisation of staining was quantified using NIS Elements Software or manual blinded counting using the Cell Counter plugin on ImageJ where indicated.

**Second-harmonic generation imaging.** Second-harmonic generation was analysed in sections of fresh frozen mouse mammary tumours and wounded skin embedded in OCT using a ZEISS 7MP multi-photon microscope. A 20×/1.0 DIC VIS-IR M27 75 mm objective immersed with water and a Coherent Chameleon Ti: Sapphire laser were used at 900 nm with SP 485 and BP 575–610 filters. Laser power and detector settings were kept constant throughout multiple sample imaging. Second-harmonic generation signal was quantified using ImageJ from representative fields of view.

**Transcriptomic and patient survival data.** Breast, lung and gastric cancer patient survival data were acquired from the Kaplan–Meier plotter[76–78]. Data have been selected and extracted using R, with Biobase and Limma libraries. Survival groups

for hi and low expression of the HWCs were selected based upon 'best cut off' for each data set (data set GSE62254 was excluded from the gastric cancer survival curves due to its extended parameters). Healing wound mRNA expression data were obtained from the NCBI's Gene Expression Omnibus with accession numbers GSE23006[28] and GSE28914[37] and from ArrayExpress with accession number E-MTAB-1323[36]. The microarray data for FAP[+] TAMs and M0 macrophages are accessible through GEO Series accession number GSE113034 [https://www.ncbi.nlm.nih.gov/geo/query/acc.cgi?acc=GSE113034].

**Statistics**. Normality and homogeneity of variance were determined using a Shapiro–Wilk normality test and an F-test, respectively. Statistical significance was then determined using a two-sided unpaired Student's t test for parametric or Mann–Whitney U-test for nonparametric data using GraphPad Prism 6 software. When comparing paired data, a paired ratio Student's t test was performed. A Welch's correction was applied when comparing groups with unequal variances. For microarray gene analysis, significance of differences (fold change) between the groups were assessed with Partek® Genomics Suite® software (Partek®) using an ANOVA test. Correction for multiple hypotheses was applied to P values by controlling the percentage of false discovery rate. Adjusted P values of <0.01 were considered significant. Statistical analysis of tumour growth curves was performed using the 'CompareGrowthCurves' function of the statmod software package[79]. No outliers were excluded from any data presented and all experiments were replicated at least twice.

**Study approval**. The use of animals for this study was approved by the Ethical Review Committee at King's College London and the Home Office, UK. Human breast adenocarcinoma tissue was obtained with informed consent under ethical approval from the King's Health Partners Cancer Biobank (REC reference 12/EE/0493).

**Data availability**. The mircroarray data that support the findings of this study are available through GEO Series accession number GSE113034. The authors declare that all other data supporting the findings of this study are available within the paper and its supplementary information files.

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

## Acknowledgements

The authors thank Mr Thomas Hayday and Dr Yasmin Haque, King's College London, for cell sorting and flow cytometry assistance, the Nikon Imaging Centre (KCL) for use of their facilities and assistance with confocal microscopy analyses, Dr Claire Mitchell (KCL) for support and assistance for second-harmonic generation (SHG) imaging, Dr Alka Saxena and Dr Paul Lavender (KCL) for useful discussion/advice/support regarding the transcriptomic analyses, and Miss Rosamond Nuamah for running the microarray. This work was funded by a grant from the European Research Council (335326). E.H. is supported by an FWO postdoctoral fellowship, and P.G. was supported by a grant from the Wellcome Trust (101529/Z/13/2). V.S.M. is supported by Cancer Research UK grants C33043/A12065 and C33043/A24478. T.M. would like to thank KCL for the George Brownlee Award, which supported this work. J.W.O. and P.K. are supported by the UK Medical Research Council (MR/N013700/1) and are KCL members of the MRC Doctoral Training Partnership in Biomedical Sciences. The research was supported by the Experimental Cancer Medicine Centre at King's College London and the National Institute for Health Research (NIHR) Biomedical Research Centre based at Guy's and St Thomas' NHS Foundation Trust and King's College London.

## Author contributions

T.M., J.C. and J.N.A. conceived the project, designed the approach, performed experiments, interpreted the data and wrote the manuscript. E.H. designed the approach, performed experiments, interpreted the data, and provided key expertise. M.O., J.W.O., P.K., M.G., P.G., S.L., D.M.K. and L.P. designed the approach, performed experiments and interpreted the data. J.D.S., C.E.G., S.S.D. V.S.-M. and T.N. designed experiments, interpreted the data and provided key expertise.

## Additional information

**Competing interests:** The authors declare no competing interests.

