## [Peer Review File · Nature Communications]

Reviewers' comments:

Reviewer #1 Expert in breast cancer metastasis:

This manuscript analyzes tumor infiltrating macrophage populations to identify a subset that expresses FAP and HO1 which happens also to be expressed in healing skin. In general, the individual experiments are performed well and the interpretation is consistent with the data shown. The findings reinforce previously published studies showing inflammatory expression profiles in tumor samples, albeit this time in enriched subsets of cells. While the authors correlate the FAP/HO1 expression to metastatic efficiency but those connections are less direct than other aspects of the data presented. So, while there are definite strengths, there are several aspects of this manuscript which raise some concerns.

1. A major concern is the use of 4T1 cells as they are unusual because of their robust MDSC response. Partly mitigating this concern is that some experiments show effects in the E0771 model and some tumors. However, the artifactual nature of the 4T1 model does require some more careful assessment of the findings before extrapolation.
2. Expanding upon point #1, E0771 is (almost as) metastatic as 4T1, but without the MDSC response. The relevance of the findings presented here are undermined by that point. A more relevant study using sister cell lines isolated from the same tumor as 4T1 (67NR, 4T10, etc.) with variable metastatic potential should be done.
3. Also a relatively major concern is that the findings in 4T1 and E0771 are observed on two completely different mouse backgrounds (BALB/c and C57BL/6, respectively). Publications from the 1970's and 1980's from several groups showed major differences in strain responses to tumors. Those studies were ignored in the citations and interpretation.
4. Which strain background was the RAG-/- mouse? Strains alter immune responses.
5. The experiments described for FAP knockout would also deplete HO1 subpopulations with FAP; therefore, the conclusions are more ambiguous than presented. The authors need to describe independence or concordance issues with FAP and HO1.
6. The studies would have been strengthened by testing behaviors in immunocompromised mice which still elicit a TAM response to orthotopic tumors.
7. While there are some molecular details presented in this manuscript, the findings are entirely conceptually similar to the published inflammation signature published by several groups (most notably Stanford).
8. The authors propose a signature to define patients more susceptible to tumor progression that they name the HWS signature (IL-1B, IL-6, SPP1). Interestingly however, their primary findings in the manuscript support IL-6 promoting these effects independently. Therefore, it is important to determine whether IL-6 can significantly identify these patients alone.
9. P8: Tumors make things that get incorporated into the microenvironment. The differences in ECM should consider this point and elaborate.
10. The authors argue that the effects of the HO-1 inhibitor are a result of macrophage specific inhibition. As the HO-1 inhibitor has the potential to target the electron transport chain it is important for the authors to fully characterize these effects, and distinguish that the effects observed in vitro and in vivo are macrophage specific. Does the HO-1 inhibitor prevent tumor migration and progression in the E0771 model?
11. Along these lines, does the HO-1 snMP inhibitor reduce the percentage of FAP+, HO-1+ TAMs in the primary tumors, or does it just inhibit their downstream signaling?
12. Studies with SnMP apparently utilize only one dose. A dose-response is required because ALL drugs have off-target effects.
13. Were all of the studies with cytokine knockout done on syngeneic backgrounds? Were all repeated in both cell lines?
14. P13: The paragraph beginning at line 328-346 is confusing and needs to be re-written for clarity. The results assessing intra- vs extra-vascularity are muddled.
15. The paragraph regarding microbiome is interesting, but very speculative based upon what is presented. It could be deleted without negatively impacting the manuscript.

16. The statistical tests used for all studies are not clearly defined. Some experiments were performed only once and were underpowered (5 mice/group).
17. Figure 1: The experiment involving injection of both E0771 and 4T1 is not interpretable as performed. A large body of literature shows tumor cell communication to distant sites (e.g., Folkman's anti-angiogenic studies). Therefore, the study needs to be done with one tumor per mouse. Moreover, if the cytokine effects on eliciting TAM are correct as the authors posit, then why doesn't the TAM population in E0771 change?
18. In Figure 1, the authors demonstrate differences in the 4T1 and E0771 tumor models, and state that the tumors were harvested at the same time. Therefore, it is important to provide the average tumor sizes for these two models upon collection. This is an important experimental consideration as well as an ethical one - i.e., was tumor volume excessive?
19. In Figure 5, based on exogenous CO driven migration changes, the authors infer that this is also the primary mechanism for TAM HO-1 mediated migration changes. However, the authors don't specifically address this question by measuring CO levels in FAP+ HO-1+ TAMs.
20. There are several statements or terms sprinkled throughout the manuscript which are inaccurate, overstatements or poorly defined.
- a. Line 50 - Macrophages are not more abundant than lymphocytes or fibroblasts in most tumors
 - b. Line 52 - "TAM collective" is not defined
 - c. Line 56 - 'ectopic' Lewis Lung adenocarcinoma is unclear what this means. Ectopic means that the cells are not in a physiologically relevant tissue. But the authors' meaning is unclear.
 - d. Line 99 - what does 'tumour's anatomical location in the mammary fat pad' mean?
 - e. It is not always clear whether tumors were placed orthotopically or subcutaneously in the text.
 - f. Line 126 - A 8 mm punch biopsy is an apparent typographical error. In the figures and M&M, the value is listed as 1 mm. Please clarify/correct.
 - g. Line 143 - What does 'terminal size' 4T1 mean?
 - h. Line 240 - The word seeding is unclear. Do the authors mean seeding only or do they imply colonization? If the former, then metastasis is not measured. If the latter, the data need to be shown with sufficient statistical power.

Reviewer #2 Expert in TAMs and the tumour microenvironment:

Ms# NCOMMS-17-23538

Journal: Nature Communication

Title: Tumour associated macrophages exploit an innate wound healing response to facilitate metastasis

Authors: Muliaditan et al. (Arnold group)

Summary

The overall goal of this study is to investigate the phenotype of FAP+ HO-1+ macrophage subpopulation, their regulation by the surrounding regenerative and/or tumor microenvironment and their roles in metastatic spreading of breast cancer cells.

The authors show that a specialized population of TAM exists in murine and human primary breast tumors, which express the molecules FAP and HO-1, specific to tumor educated TAMs.

By comparing the transcriptome of cytokines present in skin wound healing and 4T1 breast tumors, they reported similarity in the tissue response, including the emergence of FAP+ HO-1+ macrophages. Interestingly, the signature they derived using murine models and human skin wound healing response was found associated with poor prognostic different types of human cancers, and capable of inducing HO-1 expression in bone marrow derived macrophages. Il-6, one of the cytokine identified in the wound healing signature, was however insufficient to induce FAP expression in absence of the collagen-rich environment of wounded tissue or of a growing tumor. The authors showed that functional consequences of HO-1 expression in TAM is an increased transendothelial migration ability in vitro, and in vivo. Orthogonally, Il-6 regulation of FAP+ HO-1+ macrophage also affected the metastatic potential of 4T1 tumors. Thus, the authors conclude that

FAP+ HO-1+ TAMs are a potentially targetable tumor educated population of macrophages facilitating metastasis.

General comments

This study addresses the parallel between the wound healing/regenerative and breast cancer environment, with, in the latter, a focus on education of macrophage into pro-metastatic TAM expressing FAP and HO-1. Altogether, the experiments are well performed and thorough, and the figures are visually attractive. The approaches aiming to identify the regulatory processes of FAP and HO-1 expression in wound-healing skin and 4T1 TAM provide a good basis for the conclusions, and the pre-clinical trial intervention with HO-1 inhibitor is interesting. Additionally, validation of a subset of the data in human tumors, and derivation of the regenerative/wound healing-like signature strengthens the manuscript.

Nevertheless, several major concerns are raised:

1- It is unclear when the expression of FAP and HO-1 is first detected in 4T1 tumors, is there a correlation with angiogenesis or tumor size? The authors should provide additional information on the dynamic expression of these markers in TAMs as tumor progresses, as well as in metastasis. Also, whether FAP and HO-1 expression is found in particular subsets of human breast cancers should be discussed.

2- The species conserved healing wound signature (HWS) used as the basis for comparing a wounded and tumor environment has been derived from skin wound models. While interesting and supported by human data, it would be more relevant to compare breast wound healing signature to the 4T1 tumor microenvironment (TME) one. Indeed, the composition of the skin environment is very different of the breast one in homeostatic condition, with a lot of tissue resident cells. Human proteome profiling of breast cancer biopsies has also been done (Groessler et al, 2014) revealing a wound healing like signature, and could be used to compare the authors findings presented here. Additionally, it is unclear whether the skin wound signature would also then be found in skin tumors, associated with FAP and HO-1 expression, IL6 secretion, cell migratory phenotype etc... The authors should also correlate this signature to wound healing gene expression described in TAMs in the literature (which differs from M2-like/alternative activation see for review Ginhoux et al, Nat Immunology 2016 and Mantovanni et al, J.Pathol 2013). For instance, genes as Retnla, Chil3, TGFb are involved in both inflammation and wound repair, which would be important to consider here.

3- What justifies the choice of the 12 and 24h post wounding time point as representing a regenerative environment to compare with the 4T1 TME? Acute inflammation is very likely to be dominant then, not necessarily healing yet. Does then the 4T1 TME resemble an inflamed or a healing wound? This deserves clarification.

4- In the same line to the previous point, the cytokines that the authors used to define the wounded/regenerative environment, IL1b, IL6 and the Osteopontin precursor Spp1, are well known to be inflammatory mediators and chemoattractants of macrophages in injury and tumor, and it is unclear how specific of a wounded/regenerative environment they are. The authors should refine this signature, which as it is now, seems as much inflammatory as it is a wounding/regenerative one. RNA sequencing of macrophages subpopulations sorted from E0771 and 4T1 tumors, and from wounded breast models would be needed to validate the key molecules involved.

5- The IL6 gene expression (and other cytokines) levels in 4T1 tumors presented in Figure 3b is difficult to assess without the proper control of a non-tumor breast tissue, or a wounded breast mammary gland.

6- The autocrine vs paracrine role of IL-6 on TAM is insufficiently assessed/discussed, the authors should target specifically IL6 in macrophages in vivo with genetic tools to determine the effect on

HO-1 expression in TAMs.

7- The authors suggest that collagen shapes TAM response to IL6, to promote FAP expression in TAMs. Is it the presence or the fibrillar organization of collagen? What then would regulate the absence of FAP expression in collagen-rich tumors in other organs, i.e. what are the other putative regulators of FAP expression? The authors should attempt to disrupt the collagen organization and/or levels in 4T1 tumors to determine if it would affect FAP expression in TAMs. In general, the authors need to strengthen this mechanistic part of the manuscript, it is insufficient as it is to fully support their conclusions.

8- It is somewhat surprising that macrophage numbers remain unchanged during the experimental wounded skin, as multiple studies have shown the increase in macrophages in skin injury (reviewed in Rodero et al, 2010, Mahdavian Delavary et al, 2011). Have the authors looked at later time points? What is the status of HO-1 and FAP expression?

9- It is important that the authors characterize the other immune cells involved in E0771 and 4T1 tumors in a dynamic manner, to determine if the changes in FAP and HO-1 expression in TAMs is associated with recruitment of immunosuppressive adaptive cells which could also favor metastasis.

10- Knowing that HO-1 is an inducible enzyme participating in heme degradation and involved in oxidative stress resistance, are there changes in the ROS levels in the TME due to HO-1 expression in TAMs? This putative effect of HO-1+ macrophages is completely ignored in the manuscript.

Specific comments:

- In Fig. 2b and 2c, the % of F4/80+ FAP+ macrophages assessed by flow cytometry or IF is significantly different. Does that suggest that other CD45+ cells are FAP+?
- Fig. 5 would benefit from introducing a hypoxyprobe to identify hypoxic tumor areas, and needs quantitation of perivascular vs avascular associated TAMs.
- Knowing that the transendothelial migration experiments in Fig. 5 are done with matrigel and not collagen, what is the expression status of FAP in the IL6-stimulated macrophage in that system? According to results obtained in Fig.4f, these macrophages should be FAP-. How do the authors reconcile these results?
- The manuscript would benefit from a graphical abstract.
- The error bars are sometimes very large (Fig. 4k, 4j 6b, 7b), and the authors would benefit in increasing the n in these cases to strengthen their conclusions.
- In figure 6b, the mRNA expression level of FAP in TAM is reported, when it should be the % of TAMs (IF quantitation as done in previous figures).
- The effects of SnMP on cell survival (both macrophages and 4T1) or proliferation should be assessed.

Reviewer #1 Expert in breast cancer metastasis:

1. A major concern is the use of 4T1 cells as they are unusual because of their robust MDSC response. Partly mitigating this concern is that some experiments show effects in the E0771 model and some tumors. However, the artifactual nature of the 4T1 model does require some more careful assessment of the findings before extrapolation.

Our reasoning for the selection of the 4T1 model was mainly based on the observation that the TAM phenotype in these tumours was homogeneous in their expression of FAP. In LL2 tumours, FAP⁺ TAMs represented a subset (~10% of total TAMs), making resolving the polarisation signal more difficult due to the heterogeneity in the microenvironment. The 4T1 tumours as such, offered a perfect model system to dissect the signal/s which could generate FAP⁺ HO-1⁺ TAMs. There is indeed a well characterised peripheral expansion of myeloid cells in the 4T1 model. However, it should be noted that the abundance TAMs residing in 4T1 tumours is line with other subcutaneous tumour models, including the E0771. The 4T1 model has also been widely utilised within the field, and also used in the study of the metastatic cascade.

To experimentally exclude a role for MDSC's in the FAP⁺ TAM phenotype we have transduced E0771 cells to express G-CSF (the cytokine expressed by 4T1 tumour cells to elicit the peripheral myeloid expansion). E0771/G-CSF cells, when injected into mice, efficiently expand peripheral myeloid cells. However, the TAMs in E0771/G-CSF tumours did

not express FAP. This experiment is discussed in the manuscript at line 115-123, and the data is presented in Supplementary Figure 1d-g.

2. Expanding upon point #1, E0771 is (almost as) metastatic as 4T1, but without the MDSC response. The relevance of the findings presented here are undermined by that point. A more relevant study using sister cell lines isolated from the same tumor as 4T1 (67NR, 4T10, etc.) with variable metastatic potential should be done.

We thank the reviewer for this comment. In response we have conducted replicated experiments to assess the comparable metastatic potential of 4T1 and E0771 tumours. We analysed the tumours when the volumes reached approximately 1500mm³ (day 23 for 4T1 and day 29 for E0771 tumours) and quantitated the lung metastases. In a side-by-side comparison, despite both models having lung metastases, the 4T1 were considerably more metastatic than the E0771 cells. However, we should note that we have not over interpret this observation in the manuscript. This experiment is discussed line 316-318, and the data is presented in Figure 6m and Supplementary Figure 6e. We agree with the reviewer that it would also be of interest to assess the 4T1 sister lines, but we hope that with the added experiments, and the revised story it is acceptable to have focused on the 4T1 parent line in this current study.

3. Also a relatively major concern is that the findings in 4T1 and E0771 are observed on two completely different mouse backgrounds (BALB/c and C57BL/6, respectively). Publications from the 1970's and 1980's from several groups showed major differences in strain responses to tumors. Those studies were ignored in the citations and interpretation.

Our apologies for not discussing the strain as a variable more clearly, we have now added the following citation to the text to support this:

27. Wakeham J, Wang J, Xing Z. Genetically determined disparate innate and adaptive cell-mediated immune responses to pulmonary *Mycobacterium bovis* BCG infection in C57BL/6 and BALB/c mice. *Infection and immunity* **68**, 6946-6953 (2000).

We hope that the experiment which we present for the concurrent injection of 4T1 and E0771 tumours into Balb/c immunocompromised mice directly excludes the strain as a variable in our study. This has now been more clearly stated in the text, lines 123-126, which now reads:

'To exclude any strain differences in the TAM response²⁷, 4T1 and E0771 cells were concurrently injected into opposing mammary fat pads of Balb/c *Rag2*^{-/-} immunocompromised mice (syngeneic to the 4T1 cells in which FAP⁺ HO-1⁺ TAMs had been identified.'

We hope that this clarification sufficiently addresses the reviewer's concern.

4. Which strain background was the RAG^{-/-} mouse? Strains alter immune responses.

We have now clarified this in the text, lines 124-126, which now read:

'4T1 and E0771 cells were concurrently injected into opposing mammary fat pads of Balb/c *Rag2*^{-/-} immunocompromised mice (syngeneic to the 4T1 cells in which the FAP⁺ HO-1⁺ TAMs have been identified).'

5. The experiments described for FAP knockout would also deplete HO1 subpopulations with FAP; therefore, the conclusions are more ambiguous than presented. The authors need to describe independence or concordance issues with FAP and HO1.

This is an important point to our introduction, and we are sorry that this was not clear in the original manuscript. We have rewritten the passage, starting line 61, which now reads:

'Selective conditional ablation of the FAP⁺ TAM population in an immunogenic ovalbumin (OVA)-expressing LL2 tumour using Diphtheria toxin in bone marrow chimeric FAP/Diphtheria toxin receptor transgenic (DTR Tg) mice, resulted in an immunological control of tumour growth demonstrating that this macrophage subset played an important role in immune suppression^{3, 15}. FAP⁺ TAMs represented the major tumoural source of HO-1 and pharmacological inhibition of this enzyme paralleled the observations made with conditional depletion of the producing cells, suggesting that HO-1 was essential to their biological function within the tumour³.

6. The studies would have been strengthened by testing behaviors in immunocompromised mice which still elicit a TAM response to orthotopic tumors.

An experiment where we concurrently injected 4T1 and E0771 cells into immunocompromised mice is presented in Figure 1k and j. This experiment permitted us to exclude any strain variability, and the MDSC response from contributing to the FAP⁺ TAM phenotype. We had considered continuing the study in immunocompromised mice, however, we felt that the study would have greater interest to the field by characterising the TAM phenotype in the presence of a full stromal component (including T and B cells). We also feel that the therapeutic aspect of the paper (Figure 6) is further strengthened using immunocompetent mice. SnMP (the HO-1 inhibitor) has been used in the clinic for the treatment of neonatal jaundice, making the effect on reducing lung metastases, when the mice are treated with SnMP, a particularly important observation. We hope that the reviewer finds this rationale acceptable.

7. While there are some molecular details presented in this manuscript, the findings are entirely conceptually similar to the published inflammation signature published by several groups (most notably Stanford).

Our apologies, we believe that the mechanistic aspects of our study had been largely lost in the previous manuscript. In light of this comment we re-organised the presentation of the data, and flow of the story. We hope that the revised order and extensive revision to the text now more clearly emphasises the mechanistic aspects of the study. We also had not intended to overly focus on defining a signature in this study, and we agree that there are multiple examples of signatures in the literature. If the reviewer feels that we should modify

the description from 'healing wound signature' to more simply 'healing wound cytokines' we would be happy to do this.

8. The authors propose a signature to define patients more susceptible to tumor progression that they name the HWS signature (IL-1B, IL-6, SPP1). Interestingly however, their primary findings in the manuscript support IL-6 promoting these effects independently. Therefore, it is important to determine whether IL-6 can significantly identify these patients alone.

We have now included the patient survival data for IL-6 alone, which is presented in Supplementary Figure 3c. As a single cytokine, IL-6 was not prognostic to survival. However, despite being negative, we feel that this does not negatively impact on the study, as the analysis of the signature was to question whether a healing wound-like cytokine response, when expressed within the tumour, was prognostic overall. However, we agree that it is indeed important to display the IL-6 alone graph as it will be of interest to the reader at the point the study focuses on IL-6. We have addressed this in the text, line 217, which now reads:

'Despite the ability of IL-6 to regulate HO-1 expression, it should be noted that IL-6 alone did not account for the wider prognostic value of the HWS in the Kaplan-Meier Plotter dataset (Supplementary Figure 3c), however, IL-6 has been demonstrated to be associated with poor overall and disease specific survival (DSS) by others ³⁶'

9. P8: Tumors make things that get incorporated into the microenvironment. The differences in ECM should consider this point and elaborate.

We now present a biologically replicated ECM array for the 4T1 and E0771 tumours in Supplementary Figure 4b. We also present *Col1a1* mRNA expression in different healthy tissues of the mouse for a comparison to that of the tumour in Supplementary Figure 4c. The ECM is discussed in the text between lines 230-253. We hope that the reviewer feels that we have sufficiently acknowledged the characteristics of the ECM in the revised manuscript.

10. The authors argue that the effects of the HO-1 inhibitor are a result of macrophage specific inhibition. As the HO-1 inhibitor has the potential to target the electron transport chain it is important for the authors to fully characterize these effects, and distinguish that the effects observed *in vitro* and *in vivo* are macrophage specific. Does the HO-1 inhibitor prevent tumor migration and progression in the E0771 model?

Our apologies, we had not intended to imply that in the *in vivo* models that SnMP was specifically targeting the macrophages. We have reviewed all aspects of the text to ensure there are no over-statements in the description of this. We hope that the revised structure of the manuscript more clearly defines the mechanistic aspects of the system and the possible role of macrophages using the *in vitro* transendothelial migration model (both presented Figure 6).

The reviewer's comment relating to the role of HO-1 in the electron transport chain highlighted an important point which we had not discussed in the original manuscript. In response, we investigated the survival of both 4T1 cells and macrophages exposed to various concentrations of SnMP, to demonstrate that SnMP was not simply affecting the cell viability in the transendothelial migration assay to account for these observations. These experiments are presented Supplementary Figure 7c-d, and discussed in the text, lines 330-337.

We attempted to address the reviewer's comment relating to an effect of SnMP on metastases in the E0771 tumours. However, when we investigated lung metastases in the E0771 model, although present, there was significantly fewer than the 4T1 model. In most cases a single metastasis or less was record which prevented the ability to assess a role of SnMP. However, we now present the metastases counts for the E0771 and 4T1 tumours in Figure 6m.

11. Along these lines, does the HO-1 SnMP inhibitor reduce the percentage of FAP+, HO-1+ TAMs in the primary tumors, or does it just inhibit their downstream signaling?

We can confirm that the abundance of FAP⁺ TAMs was not affected by HO-1 inhibition. In response to the reviewer's comment we now present the data to support this in Figure 6e. We have also more broadly strengthened Figure 6 with additional data to support our conclusions.

12. Studies with SnMP apparently utilize only one dose. A dose-response is required because ALL drugs have off-target effects.

We agree that there is always a concern of off-target effects when using small molecule inhibitors. However, the dose of SnMP selected for the *in vivo* study was within the equivalent dose which has been given to infants in the clinic to target hepatic HO activity for the treatment of jaundice. As such, the selected dose of SnMP used in this study provides a clinical relevance to these observations. There is also a significant prior literature for the use of SnMP. The field generally regard SnMP as one of the most specific and least toxic HO-1 inhibitors. This literature has been cited in the manuscript to provide additional support to both the dose and specificity of SnMP.

For the *in vitro* assay we have now presented dose responses for SnMP, assessing tumour and macrophage viability (Supplementary Figure 7c-d). We have also presented endothelial cell layer permeability in a dose response to SnMP (Supplementary Figure 7f). We hope that these data further support our chosen dose of SnMP used in the study.

13. Were all of the studies with cytokine knockout done on syngeneic backgrounds? Were all repeated in both cell lines?

We can confirm that the background of the *Il6*^{-/-} mice were syngeneic to the 4T1 cells (Balb/c). This has been clarified in the text, line 257.

'4T1 cells were injected into syngeneic Balb/c WT and *Il6*^{-/-} mice'

As we specifically wanted to demonstrate that IL-6 was playing a role in the FAP⁺ HO-1⁺ TAM phenotype in vivo we conducted this experiment only in the 4T1 model. We are sorry that this rationale was not clear in the original manuscript. We hope that the revised story more clearly described this.

14. P13: The paragraph beginning at line 328-346 is confusing and needs to be re-written for clarity. The results assessing intra- vs extra-vasation are muddled.

In response to the reviewer's comment we have re-written the passage, lines 388-403, which now read:

'TAMs have been demonstrated to play a critical role in metastatic spread^{7, 58} and tumour cells and TAMs are closely associated during intravasation⁴⁰. A well-characterised example of this interaction is through a paracrine loop involving M-CSF released from tumour cells, and epidermal growth factor (EGF) released by TAMs⁵⁹. The importance of the M-CSF/EGF paracrine loop in metastasis has been clearly demonstrated by mammary xenograft models using a conditional knockout of CSF-1⁵⁹. We demonstrate that HO-1 expression, which is up-regulated in response to the HWS cytokine IL-6, plays a role in facilitating transendothelial migration of tumour cells, thereby supporting their metastatic spread. HO-1 expression also correlates with metastasis in human cancer⁶⁰ and has previously been implicated with metastasis in murine models⁶¹. While our results indicate that FAP⁺ HO-1⁺ TAMs facilitate tumour cell intravasation, supported by their presence in the perivascular region of the tumour, others have also described a role for lung resident HO-1⁺ macrophages in facilitating the extravasation of metastatic tumour cells entering the lung⁶¹. Despite our results suggesting that lung colonisation of 4T1 cells was HO-independent, these findings highlight the broader physiological role of HO activity in transendothelial migration, which is exploited by different tumours.'

15. The paragraph regarding microbiome is interesting, but very speculative based upon what is presented. It could be deleted without negatively impacting the manuscript.

In light of the reviewer's comment we have removed this discussion from the text.

16. The statistical tests used for all studies are not clearly defined. Some experiments were performed only once and were underpowered (5 mice/group).

We can confirm that each experiment presented in both the main body and Supplementary Material have been replicated. We have also improved the cohort sizes used in many of the experiments.

Regarding the statistical tests used, to reduce the size of the individual Figure Legends, we had elected to have a single comprehensive statistics section in the Supplementary Methods (starting line 362). However, should the reviewer feel that we need to include these tests within the Legends, we would of course be happy to revert to this.

17. Figure 1: The experiment involving injection of both E0771 and 4T1 is not interpretable as performed. A large body of literature shows tumor cell communication to distant sites (e.g., Folkman's anti-angiogenic studies). Therefore, the study needs to be done with one tumor per mouse. Moreover, if the cytokine effects on eliciting TAM are correct as the authors posit, then why doesn't the TAM population in E0771 change?

In response to the reviewer's comment, we have now presented the accompanying data for wild type mice bearing a single E0771 tumour within the main manuscript, displayed Figure 1i-j. All data for both 4T1 and E0771 (other than Figure panels 1k-l) have been conducted using syngeneic mice injected with a single tumour.

In the mice concurrently injected with 4T1 and E0771 tumours, the observation that FAP⁺ HO-1⁺ TAMs remained restricted to the 4T1 tumour model would be in agreement with our observations in Figure 4, as the macrophages would need to receive both IL-6 and engage collagen ECM (which is specific to the 4T1) to elicit the expression of FAP. However, this experiment was in part conducted to investigate the possibility of communication, and the data interestingly suggested that such a communication did not affect the FAP⁺ HO-1⁺ phenotype.

18. In Figure 1, the authors demonstrate differences in the 4T1 and E0771 tumor models, and state that the tumors were harvested at the same time. Therefore, it is important to provide the average tumor sizes for these two models upon collection. This is an important experimental consideration as well as an ethical one - i.e., was tumor volume excessive?

We thank the reviewer for highlighting this. We have now presented the average tumour sizes at the point of resection in Supplementary Figure 1h. These tumours do have different growth rates, and as such, there was a difference in the overall size at the point of resection, however as we have demonstrated that in the 4T1 tumour, the expression of FAP is stable throughout tumour progression (Figure 1f-g), and as the *Hmox1* mRNA was comparable to that of late stage tumours shown in Figure 1h (for 4T1) and 1j (for E0771), we feel that it is reasonable to conclude that the difference in size at the point of resection did not impact the validity of our conclusions.

We can confirm that the combined tumour volume of the 4T1 and E0771 tumours in these experiments did not exceed the Home Office licence limits, or that of the NCRI guidelines.

19. In Figure 5, based on exogenous CO driven migration changes, the authors infer that this is also the primary mechanism for TAM HO-1 mediated migration changes. However, the authors don't specifically address this question by measuring CO levels in FAP⁺ HO-1⁺ TAMs.

We entirely agree with the reviewer. This is indeed a question we are keen to explore. Unfortunately, as CO is a chemically inert gas, although biologically active, and as there are

no chemical reporters to directly measure its production at the cellular level, we have not yet been able to address this. However, the use of sealed chambers supplying CO₂, as used in this study, have been widely accepted as a 'gold standard' approach by the field. However, the reviewer's suggestion would indeed be an interesting aspect to resolve.

20. There are several statements or terms sprinkled throughout the manuscript which are inaccurate, overstatements or poorly defined.

We thank the reviewer for highlighting these points and we have modified the text as described:

a. Line 50 - Macrophages are not more abundant than lymphocytes or fibroblasts in most tumors

Line 51 (in the revised manuscript) now reads:

'Tumour associated macrophages (TAMs) form part of the stromal cell infiltrate into solid tumours'

b. Line 52 - "TAM collective" is not defined

Line 54 (in the revised manuscript) now reads 'TAM population'.

c. Line 56 - 'ectopic' Lewis Lung adenocarcinoma is unclear what this means. Ectopic means that the cells are not in a physiologically relevant tissue. But the authors' meaning is unclear. You have indeed correctly interpreted this. The lung derived LL2 cells had been placed subcutaneously in the hind flank of the mouse, and as such were not implanted at the same location from which they were derived. However, line 56 in the revised manuscript, the word 'ectopic' has been replaced with 'subcutaneous'.

d. Line 99 - what does 'tumour's anatomical location in the mammary fat pad' mean?

Line 109 (in the revised manuscript) now reads:

'The FAP⁺ TAM phenotype was not a biological response to the microenvironment of the mammary fat pad in which the tumour cells were injected as TAMs in another orthotopic murine model of mammary adenocarcinoma, E0771^{20,21}, grown at the same anatomical location in syngeneic C57Bl/6 mice, did not express FAP (Figure 1i) and expressed lower levels of *Hmox1* (Figure 1j).'

We hope that this revision adds clarity to the description.

e. It is not always clear whether tumors were placed orthotopically or subcutaneously in the text.

We have clarified this in the text, line 97 now reads:

'we utilised an orthotopic model of mammary adenocarcinoma in which 4T1 tumour cells are injected into the mammary fat pad of Balb/c syngeneic mice ¹⁹,

Also, in the Supplementary Methods we have clarified this, line 154 now reads:

'4T1 or E0771 cells (2.5×10^5 in 100 μ l RPMI) were orthotopically implanted by subcutaneous injection into the mammary fat pad of female mice that were six to eight weeks of age.'

f. Line 126 - A 8 mm punch biopsy is an apparent typographical error. In the figures and M&M, the value is listed as 1 mm. Please clarify/correct.

Throughout the text we have clarified our wounding experiments to be an 8 mm punch biopsy. Also, Supplementary Methods, line 120 now reads:

'Superficial wounding of the skin was performed using an 8 mm punch biopsy.'

g. Line 143 - What does 'terminal size' 4T1 mean?

Our apologies for the lack of specificity in this description. We have now removed all reference to 'terminal size' and included the day post tumour cell inoculation that tumours were excised. We also supply supporting growth curves throughout the manuscript so that the reader can clearly see the tumour sizes at the point of excision.

h. Line 240 - The word seeding is unclear. Do the authors mean seeding only or do they imply colonization? If the former, then metastasis is not measured. If the latter, the data need to be shown with sufficient statistical power.

In response to the reviewer's comment we have altered the description to 'colonisation'. We have also increased the n number for the experiment shown in Figure panel 6j to n=10.

Reviewer #2 Expert in TAMs and the tumour microenvironment:

1- It is unclear when the expression of FAP and HO-1 is first detected in 4T1 tumors, is there a correlation with angiogenesis or tumor size? The authors should provide additional information on the dynamic expression of these markers in TAMs as tumor progresses, as well as in metastasis. Also, whether FAP and HO-1 expression is found in particular subsets of human breast cancers should be discussed.

We thank the reviewer for this comment. In response, we conducted a kinetic analysis of the

4T1 tumours which is presented in Figure 1f-g and Supplementary Figure 1c. We can confirm that the FAP⁺ TAM phenotype is stable throughout tumour progression, and the phenotype can be detected at the point of tumour establishment.

Although we request to maintain the immunofluorescence data for the FAP⁺ HO-1⁺ TAMs in human breast tumours, should the reviewer feel that it is incomplete in its current form, we would be happy to remove panels Figure panels 1a and b. We feel that their removal would not negatively impact on the overall study as we cite from previous literature that FAP⁺ TAMs have previously been reported in human breast cancer.

2- The species conserved healing wound signature (HWS) used as the basis for comparing a wounded and tumor environment has been derived from skin wound models. While interesting and supported by human data, it would be more relevant to compare breast wound healing signature to the 4T1 tumor microenvironment (TME) one. Indeed, the composition of the skin environment is very different of the breast one in homeostatic condition, with a lot of tissue resident cells. Human proteome profiling of breast cancer biopsies has also been done (Groessl et al, 2014) revealing a wound healing like signature, and could be used to compare the authors findings presented here.

Although these observations of the FAP⁺ TAM phenotype in the current study have been investigated using breast cancer models, we have previously identified FAP⁺ HO-1⁺ TAMs in Lewis lung carcinoma tumours (Arnold et al., 2014). As such, we would be keen not to limit the comparison to breast tissue alone. We would request permission from the reviewer to maintain the study focus on a comparison to a generic healing wound model. We believe that the fundamental nature of this macrophage response is more cleanly demonstrated using this approach. We hope that the revised manuscript more clearly clarifies our rationale on this experimental approach.

Additionally, it is unclear whether the skin wound signature would also then be found in skin tumors, associated with FAP and HO-1 expression, IL6 secretion, cell migratory phenotype etc...

In response to the reviewer's comment we studied the murine B16 melanoma model, and assessed the prognostic value of the HWS in melanoma patients. Please see the data below. We did not identify FAP⁺ TAMs in the B16 model, however the relative absence of IL-6 is consistent with the absence of FAP expression. Also, the HWS was not prognostic in patients with melanoma. We elected to not include this Figure in the revision, however if the reviewer requests, we can of course include this data. Panels a-c have been replicated.

B16 melanoma tumours do not contain FAP⁺TAMs. (a) Growth curves of B16F10 tumours grown in syngeneic C57Bl/6 mice (n = 5). (b) Representative FAP staining of live (7AAD⁻) CD45⁺ F4/80⁺ TAMs from enzyme-dispersed B16 tumour (day 17 post injection) as assessed using flow cytometry.

Histogram represent positive staining for FAP (open black) against isotype control staining (grey shaded). (c) Tumoral mRNA expression of *Ilf6* in B16 (pink, n = 9) and 4T1 (blue, n = 6) tumours relative to the housekeeping gene *Tbp*. (d) Kaplan-Meier survival curves in breast cancer patients showing disease specific survival (DSS) with high (red) and low (black) tumour expression of HWS cytokines (n = 158 in HMW^{lo} group and n = 52 in HWS^{hi} group). Bar chart is presented as mean + s.d. *** $P < 0.001$.

The authors should also correlate this signature to wound healing gene expression described in TAMs in the literature (which differs from M2-like/alternative activation see for review Ginhoux et al, Nat Immunology 2016 and Mantovanni et al, J.Pathol 2013). For instance, genes as *Retnla*, *Chil3*, *TGFb* are involved in both inflammation and wound repair, which would be important to consider here.

In response to the reviewer's suggestion we decided to include transcriptomic data that we had acquired for the 4T1 TAMs, but had not yet published (Figure 2a-b). In the revised manuscript we also investigate the transcriptome of these cells for the expression of the genes relating to the wound response, alongside phenotypic markers of M2 and M1 macrophages, which is now presented Figure 2k, and discussed lines 183-189 in the text.

3- What justifies the choice of the 12 and 24h post wounding time point as representing a regenerative environment to compare with the 4T1 TME? Acute inflammation is very likely to be dominant then, not necessarily healing yet. Does then the 4T1 TME resemble an inflamed or a healing wound? This deserves clarification.

We thank the reviewer for their comment. We have now included a kinetic analysis of cytokine expression after wounding. The analysis is now presented in Figure 2i. 12-24hours represented the peak of the cytokine response. It would suggest, as the reviewer correctly points out, that the cytokines are associated with the inflammatory phase of the wound response. In response we have also expanded our immunofluorescence analysis of the wound to study both day 2 (inflammatory response) and day 6 (healing) post wounding in the characterisation of FAP⁺ HO-1⁺ macrophages at the wound site. Our data demonstrates that FAP⁺ HO-1⁺ macrophages appear in the wound during the inflammatory phase of the wound response and are maintained thereafter in the granulation tissue at day 6 during healing. We feel that the revised Figure 2 is now greatly strengthened in response to the reviewer's comment.

4- In the same line to the previous point, the cytokines that the authors used to define the wounded/regenerative environment, IL1b, IL6 and the Osteopontin precursor *Spp1*, are well known to be inflammatory mediators and chemoattractants of macrophages in injury and tumor, and it is unclear how specific of a wounded/regenerative environment they are. The authors should refine this signature, which as it is now, seems as much inflammatory as it is a wounding/regenerative one.

We agree with the reviewer that the cytokines we have identified are more broadly associated with inflammation. However, as these only represent a small fraction of the total possible cytokines which could be associated with inflammation, we feel that having identified these genes, which are specifically regulated in this context does suggest that they are characteristic of the inflammation associated with a healing wound response. We hope that the revised manuscript more clearly distinguishes this point.

We appreciate that 'signatures' are often referred to as a wide collection of genes, involving intracellular, membrane bound and secreted molecules, and our cytokine panel is a relative discrete gene set by comparison. Should the reviewer feel that it is required we could modify our description of these cytokines from a 'wound healing signature' to simply refer to them as 'wound healing cytokines'.

RNA sequencing of macrophages subpopulations sorted from E0771 and 4T1 tumors, and from wounded breast models would be needed to validate the key molecules involved.

In response we have now included the transcriptomic data for the 4T1 TAMs (Figure 2a-b). We use this data to demonstrate that the TAMs are utilising pathways associated with a wound healing response (Figure 2b). We also utilise these data to investigate their cytokine expression of the genes identified in the healing wound response (Figure 2k). We feel that the inclusion of this data has dramatically lifted Figure 2, and thank the reviewer for the suggestion.

5- The Il6 gene expression (and other cytokines) levels in 4T1 tumors presented in Figure 3b is difficult to assess without the proper control of a non-tumor breast tissue, or a wounded breast mammary gland.

In response we have now analysed murine mammary gland tissue for the genes of interest as a comparison to that of the tumour models. The data is now presented in Figure 2j.

6- The autocrine vs paracrine role of IL-6 on TAM is insufficiently assessed/discussed, the authors should target specifically IL6 in macrophages *in vivo* with genetic tools to determine the effect on HO-1 expression in TAMs.

We agree with the reviewer that the paracrine or autocrine signalling in this system is an interesting question. However, we have demonstrated that both neutrophils and CAFs also express IL-6 alongside the TAMs in 4T1 tumours, presented Figure 4e. These are also abundant tumour stromal cell populations, data presented Supplementary Figure 6c. As such, we feel that is highly likely that both routes of signaling do exist. We do present *in vivo* experiments conducted in the *Il6*^{-/-} mice (Figure 5), where we demonstrate that the source of the cytokine is indeed of stromal origin (as we modify the FAP⁺ HO-1⁺ phenotype in the *Il6*^{-/-} mice). However, we have ensured that in the revised manuscript we do not make a claim to either paracrine or autocrine signalling in this system.

7- The authors suggest that collagen shapes TAM response to IL6, to promote FAP expression in TAMs. Is it the presence or the fibrillar organization of collagen?

It is an interesting question, in response, the *in vitro* collagen would not be in a fibrillar network which would suggest that it is purely an engagement with this matrix that is important.

What then would regulate the absence of FAP expression in collagen-rich tumors in other organs, i.e. what are the other putative regulators of FAP expression?

We apologise that this wasn't clear in the previous manuscript. In response to the reviewer's question we believe that our data has demonstrated that the phenotype would require two signals, the first being IL-6 and the second an engagement with a collagen ECM network. We have added additional data investigating the collagen expression in a variety of normal tissues (Supplementary Figure 4c). However, as IL-6 represents at least one master regulator of the phenotype, the absence of FAP⁺ macrophages in these healthy tissues during homeostasis (Supplementary Figure 2a) could be expected due to the absence of IL-6-driven inflammation. We hope that this point is more clearly presented in the revised manuscript.

The authors should attempt to disrupt the collagen organization and/or levels in 4T1 tumors to determine if it would affect FAP expression in TAMs. In general, the authors need to strengthen this mechanistic part of the manuscript, it is insufficient as it is to fully support their conclusions.

We feel that the data in the previous manuscript had lost a lot of its mechanistic insight due to the order of presentation. We have significantly modified the structure of manuscript to highlight these points. Although, we have been unable to disrupt the collagen organisation *in vivo*, we hope that the revised story has significantly strengthened the mechanistic insights of the study which, we feel had been lost in the previous manuscript.

8- It is somewhat surprising that macrophage numbers remain unchanged during the experimental wounded skin, as multiple studies have shown the increase in macrophages in skin injury (reviewed in Rodero et al, 2010, Mahdavian Delavary et al, 2011). Have the authors looked at later time points? What is the status of HO-1 and FAP expression?

In response to the reviewer's comment we added day 6 post wounding tissue to our analysis (Figure 2). Weber et al, 2016 had demonstrated that the majority of macrophage recruitment to the wound occurred at day 3+ post wounding. In agreement with the literature we identify the accumulation of macrophages at the wound edge at day 6 post wounding (Figure 2d and f). Adding this timepoint also prompted us to consider further the location of the FAP⁺ HO-1⁺ macrophages at the wound site. This has also revealed new insight, where we have discovered that the majority of the FAP⁺ HO-1⁺ macrophages are present in the granulation tissue of the wound. In the revised manuscript we have quantitated both macrophages, and their phenotype (FAP and HO-1) within the wound edge and that of the granulation tissue (Figure 2d-g). We feel that this has greatly strengthened this aspect of the study.

9- It is important that the authors characterize the other immune cells involved in E0771 and 4T1 tumors in a dynamic manner, to determine if the changes in FAP and HO-1 expression in TAMs is associated with recruitment of immunosuppressive adaptive cells which could also favor metastasis.

In response we have now conducted the dynamic experiment for the 4T1 tumours in which we observed the FAP⁺ HO-1⁺ TAM phenotype and significant lung metastases. This data is

now presented in Supplementary Figure 6b-c. In the 4T1 model we observed an increase in neutrophils as the tumours progressed, however macrophages/monocytes still represented the largest stromal cell component in these tumours at all stages of tumour progression analysed.

10- Knowing that HO-1 is an inducible enzyme participating in heme degradation and involved in oxidative stress resistance, are there changes in the ROS levels in the TME due to HO-1 expression in TAMs? This putative effect of HO-1+ macrophages is completely ignored in the manuscript.

In response we have modified the manuscript to acknowledge the role of HO-1 in the modulation of ROS, starting line 330, which now reads:

'CO, a by-product of HO activity is a biologically active molecule that can play key roles in cytoprotection through modulating cellular signalling, the mitochondrial electron transport chain, and the generation of reactive oxygen species⁴⁸. To exclude the possibility that SnMP might have affected the viability of the 4T1 cells or macrophages, thereby reducing the number of live migrated cells, 4T1 tumour cells and macrophages were incubated with increasing doses of SnMP. However, SnMP did not affect the viability of either 4T1 cells or macrophages at any dose or timepoint tested (Supplementary Figure 7c-d). Also, neither M(IL-6) cells nor SnMP treatment affected the permeability of the endothelial monolayer (Figure 6p and Supplementary Figure 7e-f)'

Our apologies, we had undervalued the importance of the role of HO-1 in the modulation of ROS. Although not directly measuring their production we have now considered their possible role in modulating the permeability of the endothelial monolayer in the transendothelial migration assay. To address this, we now present the permeability assessment in the main manuscript, Figure 6p. We have also now included an assessment of the permeability of the endothelial layer to a dose response of SnMP, presented in Supplementary Figure 7f. Also, a modulation of ROS could affect the viability of the cells in this assay. We have now also assessed this possibility in both a time- and dose-dependent manner using SnMP. The data for this is presented in Supplementary Figures 7c and d.

Specific comments:

- In Fig. 2b and 2c, the % of F4/80+ FAP+ macrophages assessed by flow cytometry or IF is significantly different. Does that suggest that other CD45+ cells are FAP+?

We thank the reviewer for highlighting this ambiguity, and our apologies for not addressing this. We believe that our digestion procedure may have enriched for the FAP⁺ macrophage, as it was optimised for digesting soft tumour tissue. As such, in the revised manuscript we focused on quantitating the macrophages, their location and phenotype using immunofluorescence in the skin and wound. The data is now presented in Figure 2f-g. Also, as described above, we have now added day 6 post wounding to our analysis. We feel that this new presentation is far stronger and has strengthened our conclusions.

- Fig. 5 would benefit from introducing an hypoxyprobe to identify hypoxic tumor areas, and needs quantitation of perivascular vs avascular associated TAMs.

Our apologies, in the original manuscript we had discussed the published observation that HO-1 expression is induced by hypoxia. In response we have decided to remove this discussion from the text. This aspect of the study was specifically investigated to assess the location of these cells in proximity to the viable vasculature where they would have the potential to facilitate metastasis. We have now focused our discussion on this aspect alone. We hope that the reviewer finds this approach acceptable in the context of the modified discussion around this experiment.

- Knowing that the transendothelial migration experiments in Fig. 5 are done with matrigel and not collagen, what is the expression status of FAP in the IL6-stimulated macrophage in that system? According to results obtained in Fig.4f, these macrophages should be FAP-. How do the authors reconcile these results?

Indeed, these macrophages are FAP⁻ as the reviewer suggests. The FAP status of the macrophages is presented Figure 4f. We have revised the position of this experiment in the text to more clearly define our reasoning for this approach. We have also added a clarifying passage, lines 325-328 which now reads:

'To investigate whether the IL-6/HO-1 axis might directly facilitate transendothelial migration of tumour cells, BMDM were exposed to IL-6 to upregulate HO-1 expression (Figure 4a) and co-cultured with 4T1 tumour cells on an endothelial cell layer in an *in vitro* transwell assay (Figure 6n and Supplementary Figure 7a-b'

We also more clearly highlight in the discussion that we are using FAP as an *in vivo* marker of the cell type, and not attempting to investigate functionality in the current study, line 382-386, which reads:

'Although in the current study FAP has only been used as a marker, its expression could feasibly facilitate a macrophage's ability to migrate through the collagen networks found in the dermis and in the tumour microenvironment, similar to that demonstrated for FAP-expressing fibroblasts⁵⁷. FAP expression on TAMs could serve as a marker of a regenerative-like tumour microenvironment.'

We are sorry that this rationale was not clear in the previous manuscript, we hope that this is resolved in the revised manuscript.

- The manuscript would benefit from a graphical abstract.

We have now included a graphical abstract which is presented in Figure 7.

- The error bars are sometimes very large (Fig. 4k, 4j 6b, 7b), and the authors would benefit in increasing the n in these cases to strengthen their conclusions.

In response, we have now increased the n number in all the requested panels, and in places where there is a large spread of the data we have opted to present dot plots so that the distribution can more clearly be seen.

- In figure 6b, the mRNA expression level of FAP in TAM is reported, when it should be the % of TAMs (IF quantitation as done in previous figures).

Our apologies, we had mistakenly failed to define what MFI represented. We have now defined MFI 'Median Fluorescence Intensity' in the figure legend. This was the average detectable FAP protein expression on the TAM population, which was homogeneously expressed in 4T1 tumours.

- The effects of SnMP on cell survival (both macrophages and 4T1) or proliferation should be assessed.

In response we have assessed cell survival in both the macrophages and 4T1 cells using different concentrations of SnMP, and durations of exposure. These data are presented in Supplementary Figure 7c-d.

We thank you both for the constructive comments which you have made. We feel that your comments have greatly strengthened the study. We hope that you like the revised manuscript.

Reviewers' comments:

Reviewer #1 (Remarks to the Author):

The authors have responded reasonably to the prior concerns.

Reviewer #2 (Remarks to the Author):

Ms# NCOMMS-17-23538

Authors: Muliaditan et al. (Arnold group)

The revised manuscript of Mulidatian et al answered the majority of the concerns raised which significantly improved the interpretation of the presented data, the timing of the reported pro-tumorigenic/pro-metastatic TAM phenotype and associated mechanistic insights.

A few concerns remain, related to the original points raised.

1- In figure 1, we asked whether different subsets of human breast tumors display differences in FAP+HO+ macrophages. The authors didn't address this point. To clarify on their response, we do not ask for removing panels 1a and b, but to discuss whether specific breast cancer types (luminal, basal, triple negative) tumors present this phenotype.

2- Additional data presented in Figure 2 strengthen the authors analyses of FAP+HO+ macrophages through the different stages of inflammatory and wound healing response. As the authors state, the wound healing cytokine phenotype used in the study is a generic skin wound healing, which is widely accepted. From the data presented in the rebuttal (that we do not suggest to include in the manuscript), the authors showed that no wound healing response is detected in melanoma murine models, associated with no IL6 expression in that model. Additional discussion should be included as to why the signature was derived from skin and not breast wound healing models, and how the consequential changes of it are relevant to breast carcinogenesis.

3- In the microarray analyses of TAMs, 'Inflammatory response' is more pronounced than 'wound healing' GO term in the new analyses showed in Figure 2b. However, here the comparison includes splenic M0 macrophages as control, and not E0771 TAMs, which would be more relevant. This, together with the rapid upregulation and then decrease of the 'wound healing' genes showed in Figure 2i, suggest (as mentioned in the previous review comments), that inflammation is a major part of the response observed, even more than subsequent wound healing. This should be discussed more thoroughly than is currently done in the manuscript.

4- As suggested by the author, using the 'wound healing cytokines' rather than 'signature' is more appropriate knowing the limited gene set it represents.

5- The authors clarified satisfactorily the interplay between collagen/FAP expression and IL6 signaling with the added results. One point remains to be discussed though, which is the differences in the collagen organization/content showed in wound and 4T1 tumors (Figure 4h-i). Is there a correlated difference in FAP+ macrophages then dependent on collagen presence (in Figure 2d for instance)? i.e: does increased collagen presence alter macrophage phenotype in a linear manner?

Dear Reviewers,

It is with great pleasure that we present to you our re-revised manuscript entitled 'Macrophages are exploited from an innate wounding healing response to facilitate cancer metastasis' for consideration of publication in Nature Communications.

Please find below our responses to the comments on the revised manuscript.

Reviewer #2 (Remarks to the Author):

1- In figure 1, we asked whether different subsets of human breast tumors display differences in FAP+HO+ macrophages. The authors didn't address this point. To clarify on their response, we do not ask for removing panels 1a and b, but to discuss whether specific breast cancer types (luminal, basal, triple negative) tumors present this phenotype.

We have now clarified the patient cohort we analysed for the study, and supply information of the tumour types. Unfortunately, as this was a small sample size (n=3), we are unable to make conclusions of the specific abundance of FAP⁺ HO-1⁺ TAMs across the different types of breast cancer in the current study. However, we remain happy to remove these panels should the reviewer find the current presentation incomplete.

We now include indication of the specific number and type of human breast tumours analysed in the present study:

Figure legend line 806 now reads:

'(a-b) Representative images of frozen human **invasive ductal mammary carcinoma** sections stained with DAPI (nuclei; blue) and antibodies against CD11b (green), FAP (yellow) and HO-1 (red) **(representative images from n=3 tumours).**'

Supplementary Methods line 321 now reads:

'Sections of fresh frozen human breast carcinoma **(all identified as grade 3 invasive ductal carcinoma, two Basal-like and one HER2⁺)**, mouse mammary tumours or wounded skin embedded in OCT were fixed in 4% paraformaldehyde in PBS (Gibco) for 10 min at RT.'

2- Additional data presented in Figure 2 strengthen the authors analyses of FAP+HO+ macrophages through the different stages of inflammatory and wound healing response. As the authors state, the wound healing cytokine phenotype used in the study is a generic skin wound healing, which is widely accepted. From the data presented in the rebuttal (that we do not suggest to include in the manuscript), the authors showed that no wound healing response is detected in melanoma murine models, associated with no IL6 expression in that model. Additional discussion should be included as to why the signature was derived from skin and not breast wound healing models, and how the consequential changes of it are relevant to breast carcinogenesis.

We now include discussion acknowledging that other skin sites have specific finely-tuned wounding responses which could further inform on the macrophage response.

Line 382 now reads:

‘In the current study we investigated the dorsal wound, however the cytokine response can also be dictated, and fine-tuned, by the anatomical location of the wound site, such as was observed between the skin and mucosal surface ²⁸. The presence of FAP⁺ HO-1⁺ macrophages at other wound sites, and their specific role within the granulation tissue, although not covered by the current study, remain important questions to consider.’

We unfortunately could not find literature specifically comparing the cytokine response of the dorsal skin to that of the skin of the breast, but we have acknowledged the importance of this consideration. We hope that the outlined changes further clarify our rationale for this approach. To further support this, we have also modified the manuscript in the following sections:

Line 149 now reads:

‘We therefore analysed the abundance of FAP⁺ HO-1⁺ macrophages in healthy and a wounded skin of mice 2 days (acute injury inflammatory response phase) and 6 days (wound healing response phase) after full-thickness **dorsal** skin wounding by a punch biopsy ^{28, 29, 30, 31} (Figure 2c-d and Supplementary Figure 2b)’

²⁸ Chen L, Arbieva ZH, Guo S, Marucha PT, Mustoe TA, DiPietro LA. Positional differences in the wound transcriptome of skin and oral mucosa. *BMC Genomics* 2010, **11**: 471.

²⁹ Weber C, Telerman SB, Reimer AS, Sequeira I, Liakath-Ali K, Arwert EN, *et al.* Macrophage Infiltration and Alternative Activation during Wound Healing Promote MEK1-Induced Skin Carcinogenesis. *Cancer research* 2016, **76**(4): 805-817.

³⁰ Hoste E, Arwert EN, Lal R, South AP, Salas-Alanis JC, Murrell DF, *et al.* Innate sensing of microbial products promotes wound-induced skin cancer. *Nat Commun* 2015, **6**: 5932.

³¹ Parfejevs V, Debbache J, Shakhova O, Schaefer SM, Glausch M, Wegner M, *et al.* Injury-activated glial cells promote wound healing of the adult skin in mice. *Nat Commun* 2018, **9**(1): 236.

We have also included clarification and justification in line 165:

‘As we established that the FAP⁺ HO-1⁺ macrophage phenotype can be detected during reparative inflammation phase of the wound response in **the dorsal** skin, we assessed the transcriptome of a murine wound **at the same site** ²⁸ for potential secreted molecules which might direct macrophage differentiation and polarisation.’

Line 366 now reads:

'Harold's Dvorak's seminal observation that tumours and healing wounds display inherent similarity¹⁸, is now a concept which is now deeply embedded in our understanding of the stromal response in cancer⁵².'

3- In the microarray analyses of TAMs, 'Inflammatory response' is more pronounced than 'wound healing' GO term in the new analyses showed in Figure 2b. However, here the comparison includes splenic M0 macrophages as control, and not E0771 TAMs, which would be more relevant. This, together with the rapid upregulation and then decrease of the 'wound healing' genes showed in Figure 2i, suggest (as mentioned in the previous review comments), that inflammation is a major part of the response observed, even more than subsequent wound healing. This should be discussed more thoroughly than is currently done in the manuscript.

In response we have added the following passages of discussion emphasising the inflammatory aspect of the responses described:

We have now clarified day 2 post wounding as an 'inflammatory' phase of the wound healing response:

Line 150 now reads: 'We therefore analysed the abundance of FAP⁺ HO-1⁺ macrophages in healthy and a wounded skin of mice 2 days (acute injury inflammatory response phase) and 6 days (wound healing response phase).'

Line 171, we have further highlighted the early phase of the wound response as 'inflammatory' where the following passage has now been included: 'The relative peak of the acute inflammatory response of the wound coincided with the identification of FAP⁺ HO-1⁺ macrophages by confocal microscopy analysis at day 2 pw (Figure 2g).'

Line 369 now reads: '. In the present study we specifically describe cytokine genes that are associated with the inflammatory phase of a wound response, comprising *Il1B*, *IL6* and *SPP1*, which are up-regulated in the inflammatory phase of both human and mouse wounds.

Line 379 now reads: 'FAP⁺ HO-1⁺ macrophages first appear within the granulation in the initial days after wounding, during the acute inflammatory phase of the response, however their presence is maintained during the wound healing response.'

4- As suggested by the author, using the 'wound healing cytokines' rather than 'signature' is more appropriate knowing the limited gene set it represents.

In response to the reviewer's preference we have now changed the reference to 'signature' to focus on the genes as 'cytokine/s' throughout the text.

5- The authors clarified satisfactorily the interplay between collagen/FAP expression and IL6 signaling with the added results. One point remains to be discussed though, which is the differences in the collagen organization/content showed in wound and 4T1 tumors (Figure

4h-i). Is there a correlated difference in FAP+ macrophages then dependent on collagen presence (in Figure 2d for instance)? i.e: does increased collagen presence alter macrophage phenotype in a linear manner?

We thank the reviewer for this comment. In response, we have now included an analysis looking for a potential linear correlation between collagen and FAP in 4T1 tumours.

Line 253 now reads 'there was no inter-tumour correlation of the TAM's expression of FAP and tumoral *Col1a1* expression (supplementary Figure 4c).'

We have also inserted a new Supplementary Figure panel (Figure 4d) to present this data.

We thank the reviewers for their comments and continued consideration of our work. We hope that we have satisfactorily addressed their concerns in the revised manuscript.

Kind Regards

Dr James N. Arnold

King's College London Tumour Immunology Group

REVIEWERS' COMMENTS:

Reviewer #2 (Remarks to the Author):

The authors have addressed the remaining concerns and discussion points. The manuscript has been improved.